# Geometry-aware training of factorized layers in tensor Tucker format

**Emanuele Zangrando**[*]
School of Mathematics,
Gran Sasso Science Institute,
L'Aquila, Italy
emanuele.zangrando@gssi.it

**Steffen Schotthöfer**[*]
Computer Science and Mathematics Division,
Oak Ridge National Laboratory,
Oak Ridge, TN, USA
schotthoefers@ornl.gov

**Gianluca Ceruti**
Department of Mathematics,
University of Innsbruck,
Innsbruck, Austria
gianluca.ceruti@uibk.ac.at

**Jonas Kusch**
Department of Data Science,
Norwegian University of Life Sciences,
Ås, Norway
jonas.kusch@nmbu.no

**Francesco Tudisco**
School of Mathematics and Maxwell Institute,
University of Edinburgh, Edinburgh, UK;
School of Mathematics,
Gran Sasso Science Institute, L'Aquila, Italy
f.tudisco@ed.ac.uk

## Abstract

Reducing parameter redundancies in neural network architectures is crucial for achieving feasible computational and memory requirements during training and inference phases. Given its easy implementation and flexibility, one promising approach is layer factorization, which reshapes weight tensors into a matrix format and parameterizes them as the product of two small rank matrices. However, this approach typically requires an initial full-model warm-up phase, prior knowledge of a feasible rank, and it is sensitive to parameter initialization. In this work, we introduce a novel approach to train the factors of a Tucker decomposition of the weight tensors. Our training proposal proves to be optimal in locally approximating the original unfactorized dynamics independently of the initialization. Furthermore, the rank of each mode is dynamically updated during training. We provide a theoretical analysis of the algorithm, showing convergence, approximation and local descent guarantees. The method's performance is further illustrated through a variety of experiments, showing remarkable training compression rates and comparable or even better performance than the full baseline and alternative layer factorization strategies.

## 1 Introduction

The memory footprint and computational cost for inference and training are major limitations of modern deep learning architectures. A variety of techniques have been developed to address this issue, aiming to reduce model size and computational complexity. Popular approaches include

---

[*]Equal Contribution

38th Conference on Neural Information Processing Systems (NeurIPS 2024).

weight sparsification [26, 59, 30] and quantization [83, 15]. However, pruning via sparsification struggles to take advantage of GPU hardware designed for dense matrices, and it is difficult to provide error estimates on model performance when using quantization [24]. Moreover, while able to reduce resource requirements for inference, these methods struggle to achieve memory reduction during training without affecting performance. As pointed out in [20, 24], training accurate sparse or quantized neural networks from the start is particularly challenging. At the same time, the training phase of modern architectures can require several days on several hundreds of GPUs [7], requiring huge memory storage and energy consumption overheads. Thus, being able to reduce the resource demand of both inference and training while maintaining model performance is of critical importance.

Along with sparsification and quantization, layer factorization is another popular and successful model compression approach. Representing layer weights using different types of matrix and tensor factorizations can yield huge memory reduction while retaining model performance and robustness [67, 13]. A wealth of recent work has shown theoretical and experimental evidence suggesting that layer weights of over-parametrized networks tend to be low-rank [3, 5, 22] and that removing small singular values may even lead to increased model performance while dramatically reducing model size [70, 68]. One significant advantage of low-rank factorizations is that a low-parametric factorized model can be used throughout the entire training [79, 36, 68] and fine-tuning phase [33, 80]. The most direct way of doing this is by representing each layer's weight tensor $W$ as the product of small factors and then directly training each factor independently in a block-coordinate descent. In the matrix case, this boils down to imposing the rank-$r$ parametrization $W = UV^\top$ for each layer $W$, with $U, V$ rectangular matrices with only $r$ columns. The network is then trained on the set of rank-$r$ matrices $\mathcal{M}_r = \{UV^\top : U, V \text{ have } r \text{ columns}\}$, interpreting the loss as a function of $U, V$ alone. This is the approach taken also by popular fine-tuning strategies such as LoRA [32, 86, 50]. When $W$ is a higher-dimensional tensor, as in the case of convolutional kernels for example, the same approach can be implemented using different types of tensor low-rank factorizations, including canonical polyadic (CP) and Tucker formats [43, 49, 63, 71, 42, 21]. While direct training of a layer's factors is widely used in deep learning, this approach has two major disadvantages:

1. The rank $r$ of the factorization needs to be chosen a-priori, and the performance of the compressed model can highly depend on it [70, 79];
2. The training flow is highly sensitive with respect to the choice of the initialization, which may result in a high-oscillatory slow-converging loss and sub-optimal performance and may require a warm-up phase during which the full model is trained prior to the rank reduction [79, 36, 68].

Point 2, in particular, is directly related to the geometry of the constraints set. In the matrix case, it is well-known that $\mathcal{M}_r$ is a Riemannian manifold with points of very high curvature near small singular values [19]. These points give rise to stiffness and result in ill-conditioning. This intrinsic poor conditioning can be overcome by projecting the gradient flow on the tangent bundle of $\mathcal{M}_r$ as presented in [68].

In the higher-dimensional tensor case, we face the same problems. However, trying to adapt the approach for matrices to other tensor factorizations is not trivial, as it may lead to prohibitive computational costs and memory requirements scaling with the order of the tensor. Moreover, not all the tensor factorizations have the required Riemannian structure to design projections and tangent planes. This paper introduces a rank-adaptive geometry-aware training algorithm that trains factorized tensor layers in Tucker format taking advantage of the underlying Riemannian structure, yielding strictly better performance than direct factorizations and overcoming both Points 1 and 2 above. Our main contributions are:

- We design an algorithm for training tensor layers in Tucker format that is **rank-adaptive**, as the ranks of the layers are dynamically updated during training to match a desired compression rate;
- We provide theoretical guarantees of loss **descent**, **convergence** to stationary points in expectation, and **approximation** to the ideal full model;
- We provide extensive **experimental evaluation** showing that the proposed method yields remarkable training compression rates (e.g., more than $95\%$ for VGG16 on CIFAR10), while achieving comparable or even better performance than the full baseline and alternative baselines.

## 1.1 Related work

Related work on network compression methods differs structurally by the mathematical object of consideration, i.e., matrix- or tensor-valued parameter structures, as well as the type of parameter reduction. Weight pruning [28, 61, 73, 59, 81] enables parameter reduction by enforcing sparsity, i.e., zero-valued weights, whereas low-rank compression imposes parameter reduction by factorization of weight matrices [34, 49, 79, 66, 84] and tensors [44, 71, 4, 63, 42, 38, 42, 72, 35]. On top of approaches that transform tensor layers into compressed matrices [68, 34, 49, 79], different tensor decompositions have been used to compress convolutional layers. Such approaches include CP decomposition [44, 71, 4, 63], Tucker [42, 38], tensor trains [42, 72] or a combination of these [21]. Other works focus on performing efficient optimization on Stiefel manifolds to preserve orthonormality, with methods based on regularization or landing [57, 8, 13, 1], cheap parametrizations of orthogonal groups [46, 48] and Riemannian schemes [58, 67, 2]. Other methods consider only the floating point representation of the weights, e.g. [75, 25, 27, 14, 76], or a combination of the above [51]. From the algorithmic point of view, related work can be categorized into methods that compress networks entirely in a postprocessing step after full-scale training [60, 56, 44, 38, 21, 4], iterative methods where networks are pre-trained and subsequently compressed and fine-tuned [28, 34, 79], and methods that directly compress networks during training [68, 61]. As no full-scale training is needed, the latter approach offers a better potential reduction of the overall computational footprint. Only a few of these methods propose strategies for dynamically choosing the compression format during training or fine-tuning, e.g., by finding the ranks via alternating, constraint optimization in discrete [47] and discrete-continuous fashions [34]. However, both these approaches require knowledge of the full weights during training and overall are more computationally demanding than standard training. In [68], a rank-adaptive evolution of the gradient flow on a low-rank manifold was proposed to train and compress networks without using the full-weight representation; however, only for matrix-valued layers. While a direct extension of this method to Tucker tensors is possible, the resulting algorithm exhibits a prohibitive memory footprint and computational complexity. The development of rank-adaptive training methods for tensor-valued layers poses non-trivial challenges that may prevent loss descent and performance of the compressed net. For example, numerical instabilities arising from the CP decomposition during training have been observed in [44] and [63].

## 2 Geometry-aware training in Tucker format

For a tensor $W$, we write $\mathrm{Mat}_i(W)$ to denote the matrix obtained by unfolding $W$ along its $i$-th mode. The tuple $\rho = (r_1, r_2, \ldots, r_d)$ is called Tucker rank of $W$ if $r_i = \mathrm{rank}(\mathrm{Mat}_i(W))$. Every $d$-order tensor $W$ with Tucker rank $\rho = (r_1, \ldots, r_d)$ can be written in Tucker form (or Tucker decomposition) $W = C \times_{i=1}^d U_i$, entry-wise defined as

$$W(i_1, \ldots, i_d) = \sum_{\alpha_1, \ldots, \alpha_d = 1}^{r_1, \ldots, r_d} C(\alpha_1, \ldots, \alpha_d) U_1(i_1, \alpha_1) \cdots U_d(i_d, \alpha_d)$$

where $C \in \mathbb{R}^{r_1 \times \cdots \times r_d}$ is a *core tensor* of full Tucker rank $\rho = (r_1, \ldots, r_d)$ and the $U_i \in \mathbb{R}^{n_i \times r_i}$ are matrices with orthonormal columns. From this representation, we immediately see that if $W$ is represented in Tucker format, then the cost of storing $W$ and of performing linear operations with $W$ (e.g. matvecs or convolutions) is $O(r_1 \cdots r_d + n_1 r_1 + \cdots + n_d r_d)$, as opposed to the $O(n_1 \cdots n_d)$ cost required by the standard full representation. When $n_i \gg r_i$, e.g., $n_i > 1.5 r_i$, the latter is much larger than the former.

In the following, we develop a rank-adaptive algorithm that trains layers in Tucker form in a robust and efficient manner. Our derivation follows five points:

1. We introduce the dynamical low-rank approximation framework in Section 2.1, which provides gradient flows for layers in Tucker format. However, the direct use of these evolution equations to train the network will require prohibitively small learning rates due to the high curvature of the manifold of Tucker tensors.
2. We introduce a reparameterization in Theorem 2.1 that allows us to formulate robust dynamics for the reparametrized factors.
3. By integrating numerically the resulting gradient system (with e.g. SGD as explicit Euler) along with a basis augmentation step, we propose a geometry-aware rank-adaptive training strategy for

the network in Tucker format. This approach, however, requires $d + 1$ forward and backward evaluations of the network, resulting in significantly increased computational costs.

4. We show in Corollary 2.2 that computational costs can be substantially reduced by noting that the computation of the augmented basis can largely be simplified. This leads to our proposed training scheme in Algorithm 1. The algorithm is equivalent to the integration of the gradient system but requires only two instead of $d + 1$ gradient evaluations.

5. Due to its equivalence to the approach constructed in point 3, we can show that Algorithm 1 indirectly updates weights along straight lines on the manifold, thus leading to three main theoretical properties: loss descent (Theorem 3.1), convergence in expectation (Theorem 3.2), and a robust bound showing approximation of the full model (Theorem 3.3).

## 2.1 Dynamical low-rank approximation

For $\rho = (r_1, \ldots, r_d)$, the set

$$\mathcal{M}_\rho = \{W : \mathrm{rank}(\mathrm{Mat}_i(W)) = r_i, \ i = 1, \ldots, d\}$$

is a manifold with the following tangent space at any point $W = C \times_{i=1}^d U_i \in \mathcal{M}_\rho$ [40]

$$T_W \mathcal{M}_\rho = \left\{ \delta C \underset{i=1}{\overset{d}{\times}} U_i + \sum_{j=1}^d C \times_j \delta U_j \times_{k \neq j} U_k : \delta C \in \mathbb{R}^{r_1 \times \cdots \times r_d}, \ \delta U_j \in T_{U_j} \mathcal{S}_j \right\} \quad (1)$$

where $\mathcal{S}_j$ is the Stiefel manifold of real $n_i \times r_i$ matrices with orthonormal columns. To design a strategy that computes layer weights within $\mathcal{M}_\rho$ using only the low-rank Tucker factors $C$ and $\{U_i\}_i$, we formulate the training problem as a continuous-time gradient flow projected onto the tangent space (1). As shown in Section 3, the continuous formulation will allow us to derive a modified backpropagation pass which uses only the individual small factors $C, \{U_i\}_i$ and that does not suffer from a slow convergence rate due to potential ill-conditioned tensor modes (see also Section 4.2).

Let $f$ be a neural network and let $W$ be a weight tensor within $f$. Consider the problem of minimizing the loss function $\mathcal{L}$ with respect to just $W$ while keeping the other parameters fixed. This problem can be equivalently formulated as the gradient flow

$$\dot{W}(t) = -\nabla_W \mathcal{L}(W(t)) \quad (2)$$

where, for simplicity, we write the loss as a function of only $W$ and where "dot" denotes the time derivative. When $t \to \infty$, the solution of (2) approaches the desired minimizer. Now, suppose we parametrize each tensor layer in a time-dependent Tucker form $W(t) = C(t) \times_{i=1}^d U_i(t) \in \mathcal{M}_\rho$. Then $\dot{W}(t) \in T_{W(t)} \mathcal{M}_\rho$, the tangent space of $\mathcal{M}_\rho$ at $W(t)$. Thus, (2) boils down to

$$\dot{W}(t) = -P(W(t)) \nabla_W \mathcal{L}(W(t)) \quad (3)$$

where $P(W)$ denotes the orthogonal projection onto $T_W \mathcal{M}_\rho$. Using standard derivations from dynamical model order reduction literature [40], the projected gradient flow in (3) leads to the following evolution equations for the individual factors $C(t)$ and $U_i(t)$

$$\begin{aligned} \dot{U}_i &= -(I - U_i U_i^\top) \mathrm{Mat}_i\big(\nabla_W \mathcal{L}(W) \times_{j \neq i} U_j^\top\big) \mathrm{Mat}_i(C)^\dagger \\ \dot{C} &= -\nabla_W \mathcal{L}(W) \times_{j=1}^d U_j^\top , \end{aligned} \quad (4)$$

where $\dagger$ denotes the pseudoinverse and where we omitted the dependence on $t$ for brevity. Even though (4) describes the dynamics of the individual factors, the equations for each factor are not fully decoupled. A direct integration of (4) would still require taping the gradients $\nabla_W \mathcal{L}$ with respect to the full convolutional kernel $W$. Moreover, the pseudoinverse of the matrices $\mathrm{Mat}_i(C)^\dagger$ adds a stiffness term to the differential equation, making its numerical integration unstable. The presence of this stiff term is actually due to the intrinsic high-curvature of the manifold $\mathcal{M}_\rho$ and is well understood in the dynamic model order reduction community [39, 53, 37, 54, 10, 9]. As observed in [68], an analogous term arises when looking at low-rank matrix parameterizations, and it is responsible for the issue of slow convergence of low-rank matrix training methods, which is observed in [79, 36, 68].

To overcome these issues, we use the following change of variables. Let $\mathrm{Mat}_i(C)^\top = Q_i S_i^\top$ be the QR decomposition of $\mathrm{Mat}_i(C)^\top$. Note that $S_i$ is a small square invertible matrix of size $r_i \times r_i$. Then, the matrix $K_i = U_i S_i$ has the same size as $U_i$ and spans the same vector space. However, the following key result holds for $K_i$.

**Theorem 2.1.** *Let $W = C \times_{i=1}^d U_i \in \mathcal{M}_\rho$ be such that (3) holds. Let $\mathrm{Mat}_i(C)^\top = Q_i S_i^\top$ be the QR decomposition of $\mathrm{Mat}_i(C)^\top$ and let $K_i = U_i S_i$. Then,*

$$\dot{K}_i = -\nabla_{K_i} \mathcal{L}\big(\mathrm{Ten}_i(Q_i^\top) \times_{j \neq i} U_j \times_i K_i\big),$$
$$\dot{C} = -\nabla_C \mathcal{L}(C \times_{j=1}^d U_j) \tag{5}$$

*where $\mathrm{Ten}_i$ denotes "tensorization along mode $i$", i.e. the inverse reshaping operation of $\mathrm{Mat}_i$.*

The supplementary material provides the proof in Appendix D. The theorem above allows us to simplify (4), obtaining a gradient flow that only depends on the small matrices $K_i$ and the small core tensor $C$. Moreover, it eliminates a stiffness term; this added regularity appears reasonable as no inversion is now involved in the differential equations. A rigorous regularity statement can be found in Theorem 3.3. We would like to underline the importance of the careful construction of $K_i$ to arrive at this theorem. A naive extension of [68] to Tucker tensors can be constructed by a reshaping of $W$ into matrices $\mathrm{Mat}_i(W) = U_i S_i V_i^\top$ with $S_i = \mathrm{Mat}_i(C)$ and $V_i = \bigotimes_{j \neq i} U_j$. Then, $K_i = U_i S_i$ can be used to update $U_i$ into all directions $i \leq d$ which directly inherits the robustness properties presented in [68]. However, this construction of $K$ yields a prohibitive memory footprint of $\mathcal{O}(n_i \Pi_{j \neq i} r_j)$ and computational costs of $O(n_i \Pi_{j \neq i} r_j^2)$ rendering the resulting method impractical.

Based on the numerical integration of (5), we propose a robust, memory-efficient, and rank-adaptive method to update the network parameters by using only the core tensor $C$ and the basis matrices $K_i$. The proposed approach is as follows: Fix an approximation tolerance $\tau > 0$, then, first update the basis matrices performing for all $i = 1, \ldots, d$:

1. Form $K_i = U_i S_i$, where $S_i$ is the square $r_i \times r_i$ matrix from QR decomposition of $\mathrm{Mat}_i(C)^\top$
2. Compute $K_i^{\mathrm{new}}$ with one step integration of (5) starting from $K_i$
3. Form the new augmented matrix $\widehat{U}_i^{\mathrm{new}}$ by orthonormalizing the columns of $[U_i, K_i^{\mathrm{new}}]$

Second, update the core tensor and truncate:

4. Lift the core tensor $\widetilde{C} = C \times_{i=1}^d (U_i^{\mathrm{new}})^\top U_i$ using the new augmented basis matrices
5. Compute $C^{\mathrm{new}}$ with one step integration of (5) starting from $\widetilde{C}$ using fixed basis matrices $U_i^{\mathrm{new}}$
6. Perform a rank adjustment step to the prescribed tolerance by computing the best Tucker approximation of $C^{\mathrm{new}}$, i.e. solving the following optimization (rounding) task:

   Find $C \in \mathcal{M}_{\leq 2\rho}$ of smallest rank $\rho' = (r_1', \ldots, r_d')$ such that $\quad \|C^{\mathrm{new}} - C\| \leq \tau \|C^{\mathrm{new}}\|, \quad$ (6)

   where $\rho = (r_1, \ldots, r_d)$ and $\mathcal{M}_{\leq 2\rho}$ denotes the set of tensors with component-wise Tucker rank lower than $2\rho$.

In practice, the final rank adaptive step is done by performing a high-order SVD (HOSVD) [16] on $C^{\mathrm{new}}$. The parameter $\tau$ is responsible for the compression rate of the method, as larger values of $\tau$ yield smaller Tucker ranks and thus higher parameter reduction. The computed tensor $C \in \mathcal{M}_{\rho'}$ has the form $C = C' \times_{i=1}^d U_i' \in \mathcal{M}_{\rho'}$ and the computed $U_i' \in \mathbb{R}^{2r_i \times r_i'}$ with $r_i' \leq 2r_i$ are then pulled back to the initial dimension by the change of basis $U_i = U_i^{\mathrm{new}} U_i' \in \mathbb{R}^{n_i \times r_i'}$, and the new core tensor $C$ is then assigned $C'$. This implementation, however, comes at the expense of evaluating the network and gradient tapes $d + 1$ times for an order $d$ tensor.

The next key result will overcome this issue and will allow us to reduce the necessary network and gradient tape evaluations to two.

**Corollary 2.2.** *Let $W = C \times_{i=1}^d U_i \in \mathcal{M}_\rho$ be such that (6) holds. Let $\mathrm{Mat}_i(C)^\top = Q_i S_i^\top$ be the QR decomposition of $\mathrm{Mat}_i(C)^\top$ and let $K_i = U_i S_i$. Then,*

$$\mathrm{span}\left(\left[U_i, \dot{K}_i\right]\right) = \mathrm{span}\left(\left[U_i, \nabla_{U_i} \mathcal{L}(W)\right]\right).$$

The supplementary material provides the proof in Appendix E. Using Corollary 2.2, one can replace the individual forward evaluation and descend steps for $K_i$ by a single backpropagation. All available new information is given by the gradients $\nabla_{U_i} \mathcal{L}$, which can be evaluated from the same tape. Combining the above strategy with Corollary 2.2 we obtain Algorithm 1: an efficient rank-adaptive geometry-aware training method for tensor in Tucker format. Note that without the explicit computation of $K_i = U_i S_i$ we compute all basis gradients $\nabla_{U_i} \mathcal{L}$ in a single network evaluation and we use $\nabla_{U_i} \mathcal{L}$ to augment the basis. Note that stochastic gradient evaluations can be done in practice and that

---

**Algorithm 1: TDLRT**: Efficient Tensor Dynamical Low-Rank Training in Tucker format.

**Input :** Initial low-rank factors $C \sim r_1 \times \cdots \times r_d$; $U_i \sim n_i \times r_i$;
      `adaptive`: Boolean flag that decides whether or not to dynamically update the ranks;
      $\tau$: singular value threshold for the adaptive procedure.

**1** $G_i \leftarrow \nabla_{U_i} \mathcal{L}(C \times_{i=1}^d U_i)$                                 `/* Single-sweep grad evaluation */`
**2** **for** *each mode $i$* **do**
**3**    |  $U_i^{\text{new}}, \_ \leftarrow \text{QR}([U_i, G_i])$                `/* Augmentation and orthonormalization */`
**4**    $\widetilde{C} \leftarrow C \times_{i=1}^d (U_i^{\text{new}})^\top U_i$         `/* Projection of Tucker core onto new basis */`
**5**    $C \leftarrow$ descent step with direction $\nabla_C \mathcal{L}(\widetilde{C} \times_{i=1}^d U_i^{\text{new}})$

**6** $(C, U_1, \ldots, U_d) \leftarrow$ Tucker decomposition of $C$ up to relative error $\tau$ as in (6)
**7** $U_i \leftarrow U_i^{\text{new}} U_i$, for $i = 1, \ldots, d$                           `/* Rank adjustment */`

---

momentum methods are applicable for the descent step on line 5 of Algorithm 1. In case the rank decreases after the retraction step, we only use the corresponding subset of the old basis functions to form the momentum term. In case of rank increase, the momentum term of the new basis vectors is initialized as zero. As a side note, the rank truncation proposed in Algorithm 1 allows to maintain the nice theoretical guarantees of the algorithm, but in practice any tensor-completion/factorization method such as [23] could be used for this step.

## 2.2 Computational Complexity

The computational costs for the full network training come from back and forward passes through each layer. For a layer with weight tensor $W \in \mathbb{R}^{n_1 \times \cdots \times n_d}$, they require $\mathcal{O}(b \prod_i n_i)$ operations, where $b$ is the batch size. When using TDLRT, these computational costs reduce to $\mathcal{O}(b \prod_i r_i + b \sum_i n_i r_i)$ operations, yielding a significant reduction in computational costs to determine the gradient. However, performing low-rank updates also adds computational costs due to several factorizations. Here, the QR and SVD on $\text{Mat}_i(C)$, which are needed in the updates of $U_i$ and the truncation step, require $\mathcal{O}(\sum_i r_i \prod_j r_j)$, and the QR on $K_i$ requires $\mathcal{O}(\sum_i n_i r_i^2)$ operations. Hence, in total, we have for every layer a cost of $\mathcal{O}(b \prod_i r_i + b \sum_i n_i r_i + \sum_{i=1}^d (n_i r_i^2 + r_i \prod_{j=1}^d r_j))$ operations for TDLRT, vs. the $\mathcal{O}(b \prod_i n_i)$ required by the full baseline. Thus, TDLRT scales linearly with the dimensions $n_i$, and for $r_i \ll n_i$, which is typically the case, see Appendix C.2, it has advantageous computational cost.

## 3 Convergence and approximation analysis

In this section, we present our main theoretical results. First, we show Algorithm 1 guarantees descent of the training loss, provided the compression tolerance is not too large. Second, we show that when Algorithm 1 is implemented with SGD with a decaying learning rate, the method converges to a stationary point in expectation. Third, we prove that the compressed network computed via the rank-adaptive TDLRT scheme in Algorithm 1 well-approximates the full model that one would obtain by standard training, provided the gradient flow of the loss is, at each step, approximately low-rank. The latter result shows that if a high-performing subnetwork of low Tucker rank exists, then the proposed TLDRT will probably approximate it. For brevity, some statements here are formulated informally, and all proofs and details are deferred to Appendix F in the SM.
Suppose that for each convolution $W$, the gradient $\nabla_W \mathcal{L}$, as a function of $W$, is locally bounded and Lipschitz, i.e., $\|\nabla_W \mathcal{L}(Y)\| \leq L_1$ and $\|\nabla_W \mathcal{L}(Y_1) - \nabla_W \mathcal{L}(Y_2)\| \leq L_2 \|Y_1 - Y_2\|$ around $W$. Then,

**Theorem 3.1** (Descent). *Let $W(\lambda) = C \times_{j=1}^d U_j$ be the Tucker low-rank tensor obtained after one training iteration using Algorithm 1 and let $W(0)$ be the previous point. Assuming the one-step integration from 0 to $\lambda$ is done exactly, it holds $\mathcal{L}_W(W(\lambda)) \leq \mathcal{L}_W(W(0)) - \alpha\lambda + \beta\tau$, where $\alpha, \beta > 0$ are constants independent of $\lambda$ and $\tau$, and where $\mathcal{L}_W$ denotes the loss as a function of only $W$.*

We now prove that the rank-adaptive training method in Algorithm 1 converges to a stationary point in expectation if implemented with SGD and decaying learning rate.

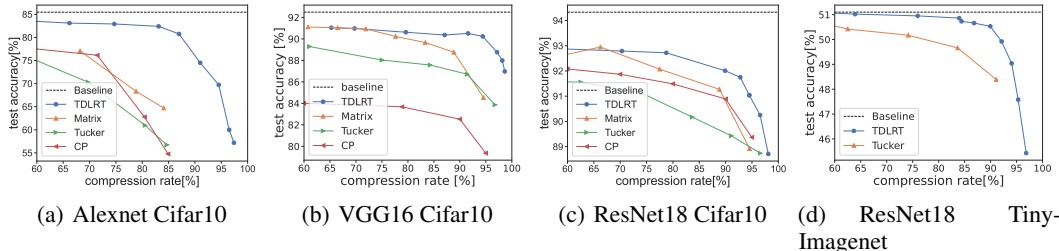

|          |          |          | (d) ResNet18 Tiny-|
| (a) Alexnet Cifar10 | (b) VGG16 Cifar10 | (c) ResNet18 Cifar10 | Imagenet |

Figure 1: Comparison of compression performance for different models against the full baseline for the Cifar10 (a-c) and Tiny-Imagenet (d) benchmark. The mean accuracy of 20 weight initializations is displayed. TDLRT achieves higher compression rates at higher accuracy with lower variance between initializations.

**Theorem 3.2** (Convergence). *Denote with $\widetilde{W}(t)$ is the weight tensor after $t$ passes of Algorithm 1 before the rank truncation step, and $W(t)$ the one obtained after the rank truncation. Assume Algorithm 1 is implemented using SGD as a descent method with learning rate sequence $\{\lambda_t\}$ satisfying the Robbins-Monro conditions:*

$$\sum_t \lambda_t = +\infty \qquad \sum_t \lambda_t^2 < +\infty \,.$$

*Suppose also that the spectral distribution stabilizes fast enough over time, i.e.,*

$$\sum_{t \geq 0} \mathbb{E}[\|\widetilde{W}(t) - W(t)\|] < +\infty$$

*and that the projected stochastic gradient has a controlled drift, namely*

$$\mathbb{E}\big[\|\nabla\mathcal{L}(t-1) \times_j P_{\widetilde{U}_j(t)}\|^2 \,|\, t-1\big] \leq \mu + \nu\|\nabla\mathcal{L}(t-1) \times_j P_{U_j(t-1)}\|^2$$

*for some $\mu, \nu > 0$, where $P_U$ is the orthogonal projection onto the range of $U$. Then, the following convergence condition holds*

$$\liminf_{t \to \infty} \mathbb{E}\|\nabla\mathcal{L}(t-1) \times_j P_{U_j(t-1)}\|^2 = 0$$

Details of the proof are contained in the appendix.

**Theorem 3.3.** *For an integer $k$, let $t = k\lambda$, and let $W(t)$ be the full convolutional kernel, solution of (2) at time $t$. Let $C(t), \{U_i(t)\}_i$ be the Tucker core and factors computed after $k$ training steps with Algorithm 2, where the one-step integration from 0 to $\lambda$ is done exactly. Finally, assume that for any $Y$ in a neighborhood of $W(t)$, the gradient flow $-\nabla\mathcal{L}_W(Y)$ is "$\varepsilon$-close" to $T_Y \mathcal{M}_\rho$. Then,*

$$\|W(t) - C(t) \times_{j=1}^d U_j(t)\| \leq c_1\varepsilon + c_2\lambda + c_3\tau/\lambda \tag{7}$$

*where the constants $c_1$, $c_2$, $c_3$ depend only on $L_1$ and $L_2$.*

In particular, both bounds in the above theorems do not depend on the higher-order singular values of the exact nor the approximate solution, which shows that the method does not suffer instability and slow convergence rate due to potential ill-conditioning (small higher-order singular values). Note that this result is crucial for efficient training on the low-rank manifold and is not shared by direct gradient descent training approaches, as we will numerically demonstrate in the following section. Moreover, we emphasize that (7) provides a sufficient condition for the computation of a high-performing subnetwork. In fact, for smooth enough network models $f$, condition (7) implies that $f(W(t)) \approx f(C(t) \times_{j=1}^d U_j(t))$, i.e. the computed subnetwork approximates the full model.

## 4 Experiments

In the following, we conduct a series of experiments to evaluate the performance of the proposed method as compared to the full model and to standard layer factorization and model pruning baselines. The full baseline is the network trained via standard implementation. In order to test the method on tensor layers, we consider here convolutional networks and apply the decomposition to the

convolutional kernel $W$. In terms of layer factorization, we compare against different baseline approaches: direct training of the factors in the low-rank matrix factorization format [34, 49, 79, 36], the low-rank tensor Canonic-Polyadic format [44, 71, 4, 63], the low-rank tensor Tucker format [42, 38], and the low-rank Tensor-Train format [62]. The convolution operation can then be written completely in terms of the small factors using the factorization ansatz. While the above papers propose different initialization and rank selection strategies, all the referenced literature trains the factors in the chosen layer factorization format by implementing forward and backward propagations simultaneously and independently on each factor in a block-coordinate fashion. This way of training on the low-rank manifold ignores the geometry of the manifold, whereas TDLRT directly exploits the underlying geometry to avoid points of high curvature. We compare the training strategy alone. Thus, we ignore any model-specific initialization and regularization addition. We also compare with Riemannian gradient descent (RGD) for tensors in Tucker format implemented using the HOSVD retraction [74, 17] and with the matrix dynamical training algorithm [68], where the standard forward and backward passes are endowed with a rank-adaptive QR projection step, similar to the proposed Algorithm 1. In terms of pruning techniques based on sparsification, we compare with methods from two of the most popular strategies: iterative magnitude pruning (IMP) [20], and single-shot pruning at initialization, single-shot network pruning (SNIP) [45] and Gradient Signal Preservation (GraSP) [78]. The experiments are performed on an Nvidia RTX3090, Nvidia RTX3070 and one Nvidia A100 80GB. The code is available in the supplementary material.

## 4.1  Compression Performance

The compression performance of TDLRT is evaluated on CIFAR10 and tiny-imagenet. The typical data augmentation procedure is employed for this dataset: a composition of standardization, random cropping, and a random horizontal flip. All methods are trained using a batch size of $128$ for $70$ epochs each, as done in [79, 36]. All the baseline methods are trained with the SGD optimizer; the starting learning rate of $0.05$ is reduced by a factor of $10$ on plateaus, and momentum is chosen as $0.1$ for all layers. The rank $\hat{r}$ of each tensor mode for the fixed-rank baseline methods is determined by a parameter $\kappa$, i.e., we set $\hat{r} = \kappa r_{\max}$. The proposed TDLRT method employs Algorithm 2, where SGD is used for the descent steps at lines 4 and 9, with momentum and learning rate as above. Dynamic compression during training is governed by the singular value threshold $\tau$, see Equation (6).

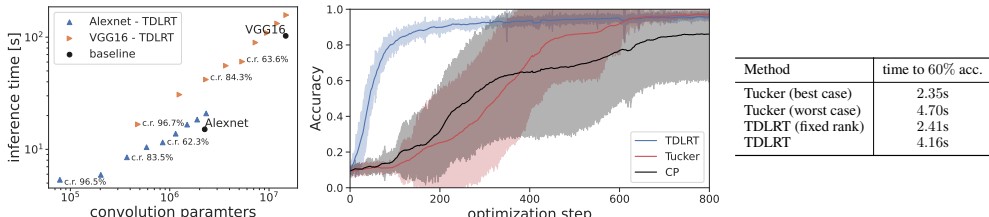

Figure 2: Left panel: Computational footprint of low-rank convolutions. TDLRT surpasses the baseline performance for meaningful compression rates. Middle panel: Convergence behavior of Lenet5 on MNIST dataset in the case of an initial overestimation of the rank, with exponentially decaying singular values. Mean and standard deviation (shaded area) over $10$ random initializations. Right panel: Compared to standard Tucker decomposition, with and without rank adaption, wall training time to reach $60\%$ accuracy for TDLRT.

Figure 1 (a-c) shows the mean accuracy of TDLRT as compared to competing factorization baselines. TDLRT achieves higher compression rates at higher accuracy with lower variance between weight initializations than the competing methods. In the case of the VGG16 benchmark, TDLRT is able to maintain baseline accuracy for compression rates over $90\%$ and exceeds the baseline on average for $\tau = 0.03$, i.e., $95.3\%$ compression. Alexnet has $16.8\%$ of the parameters of VGG16. Thus, compression is naturally more challenging to achieve. Nevertheless, TDLRT outperforms the baselines and remains close to the full network performance. Similar behavior is observed on ResNet18.

Table 1 shows a comparison of the best-performing compression between all the factorization-based and pruning-based baseline methods as well as TDLRT in the CIFAR10 benchmark for Alexnet, ResNet18, and VGG16. In the proposed evaluation, TDLRT is on par or outperforms all the alternatives, including pruning based on sparsity (implemented without warmup for the sake of a fair

Table 1: Comparison of the best-performing compression rates for different methods on the CIFAR10 benchmark with Alexnet, VGG16, and ResNet18. For each column, we report first and second best results.

| | | VGG16 | | Alexnet | | ResNet18 | |
|---|---|---|---|---|---|---|---|
| | | test acc. [%] | c.r. [%] | test acc. [%] | c.r. [%] | test acc. [%] | c.r. [%] |
| | Baseline | 92.01 | 0.0 | 85.46 | 0.0 | 94.33 | 0.0 |
| Factorization | **TDLRT** (ours) | **90.23** | **94.40** | **82.39** | 83.12 | 92.72 | 78.73 |
| | Matrix DLRT [68] | 89.13 | 83.22 | 73.57 | 71.57 | 80.98 | 56.85 |
| | Tucker-factorized [38] | 86.71 | 91.4 | 70.30 | 69.74 | 91.11 | 74.19 |
| | Matrix-factorized [79] | 84.54 | 94.34 | 77.07 | 68.20 | 92.07 | 77.49 |
| | CP-factorized [44] | 82.53 | 89.98 | 76.14 | 71.46 | 91.87 | 69.95 |
| | Tucker RGD [74] | 81.48 | 84.26 | 73.88 | 74.01 | **92.76** | 74.18 |
| | TT-factorized [62] | 87.27 | 90.30 | 78.13 | **88.14** | 87.13 | 81.24 |
| Pruning | SNIP [45] | 89.58 | 56.23 | – | – | 89.50 | 78.50 |
| | IMP [20] | 87.21 | 58.54 | – | – | 90.50 | **82.50** |
| | GraSP [78] | 88.50 | 77.30 | – | – | 89.40 | 77.90 |

comparison), Tensor-Train (TT) and Tucker factorization, Riemmanian gradient descend (RGD) for Tucker decompositions, as well as the matrix-valued DLRT, due to the higher flexibility of the Tucker format where compression along each tensor mode individually is possible. The compression rate (c.r.) is computed as $1 - c/f$, where $c$ is the number of convolutional parameters in the compressed model after training and $f$ is the number of convolutional parameters of the full model. While this is the compression rate after training, we emphasize that methods based on factorizations yield an analogous compression rate during the entire training process. We also remark that no DLRT version of CP decomposition is shown as CP is not suited for dynamical low-rank training due to its lack of a manifold structure. Similar results are obtained for the tiny-imagenet benchmark, see Fig. 1 (d) and Table 3. The rank evolution of the networks during training is discussed in Appendix C.2.

## 4.2 Robustness of the Optimization

To further highlight the advantages of Algorithm 2 as compared to standard simultaneous gradient descent on the factors of the decomposition, we show in Figure 2 the accuracy history of LeNet5 on MNIST using TDLRT as compared to standard training on Tucker and CP decompositions. In the case of TDLRT, an optimization step denotes the evaluation of Algorithm 2 for all convolutional layers for one batch of training data, while for the other methods, we refer to a standard SGD batch update for all factors of the tensor decompositions of all layers. All linear layers of the network are trained with a traditional gradient descent update and are not compressed. In this experiment, we initialize the network weights to simulate a scenario where the rank is overestimated. To this end, we employ spectral initialization with singular values decaying exponentially with powers of ten. Integrating the low-rank gradient flow with the TDLRT Algorithm 2 leads to faster and more robust convergence rates of the network training process.

## 4.3 Computational Performance

The computational performance in inference and training of convolutional layers in Tucker decomposition depends on their current tensor ranks, see Section 2. We evaluate the inference time of $120K$ RGB images and memory footprint of VGG and AlexNet in Tucker factorization as used in Algorithm 2 and compare them to the non-factorized baseline models in Figure 2. As a result, for realistic compression rates, see also Figure 1, the computational footprint of TDLRT is significantly lower than the corresponding baseline model.
Rank adaptive TDLRT training comes at the additional expense of QR and SVD operations per optimization step compared to standard fixed rank training without orthonormalization. However, the increased robustness of the optimization and faster convergence reduces the computational overhead of TDLRT as demonstrated in Figure 2. In order to provide a fair comparison between the methods, we report the time to achieve $60\%$ accuracy target at a compression rate of $90\%$ for all methods, on

LeNet5 MNIST with the setting as in Section 4.2. We refrain from measuring time to convergence since the standard Tucker decomposition is not able to reach similar accuracy levels at the same compression ratio as TDLRT.

## 4.4 Fine-tuning with LoRA-like low-rank adapters

In this section, we are presenting another application of our method, namely fine-tuning pre-trained models through the use of low-rank adapters. In particular, our approach Algorithm 1 is completely agnostic between model compression and adaptation, the approach is the same. More precisely, given a pre-trained model $f_{W^*}$ with tensor pre-trained weights $W^*$, it is possible to add a low-rank Tucker correction $W$. Then, given a task loss $L(f)$, by defining $\mathcal{L}(W) = L(f_{W^*+W})$ we report ourselves to the original formulation of the problem. Moreover, we would like to stress that this approach can be applied also to matrices, which are a particular case of tensors with $d = 2$ modes. To showcase these two settings, we present two different settings in which we test our method. In Table 2 (left) we show the fine-tuning of Deberta V3 [29] on the GLUE benchmark [77]. In this test case here, the low-rank adapters have been applied to all matrices in attention layers, and the final performance against LoRA [33] fine-tuning is reported. Apart from our additional hyperparameter $\tau$, all the other hyperparameters had been kept as reported in [85].

In Table 2 (right), we report the results for fine-tuning stable diffusion [64] with Dreambooth [65]. To adapt the model, we applied Tucker tensor corrections to each convolution of the Unet, and a matrix adapter to each attention layer of the text encoder network. We applied our method on all these correctors, while keeping the same hyperparameters setting as in [55]. We would like to observe that the kind of adapters they propose for convolutions consist in a matrix factorization of a reshaping of the convolutional tensor. This results in a potentially bigger number of parameters needed, as a matrix factorization would be $O((Cd_1d_2 + F)r)$ against a $O(Cr_1 + Fr_2 + d_1r_3 + d_2r_4 + r_1r_2r_3r_4)$ for a plain Tucker factorization, where $F$ is the number of output features, $C$ is the number of input channels, $d_1$ and $d_2$ are the spatial dimensions of the convolutional kernel. This observation is in fact reflected in the numbers in Table 2, in which we can observe a higher compression potential for tensor decompositions compared to matrix ones.

Table 2: Fine-tuning performance metrics on Deberta V3 Glue benchmark (left) and on Stable diffusion Dreambooth (right).

| GLUE | LoRA | TDRLT(Ours) |
|---|---|---|
| # params | 1.33M (rank 8) | 0.9M ($\tau = 0.15$) |
| CoLa (Corr.) | 0.6759 | 0.7065 |
| MRPC (Acc.) | 0.8971 | 0.9052 |
| QQP (Acc.) | 0.9131 | 0.9215 |
| RTE (Acc.) | 0.8535 | 0.8713 |
| SST2 (Acc.) | 0.9484 | 0.9594 |

| method | loss | # params |
|---|---|---|
| LoRA ($r = 8$) | 0.260 | 5 M |
| LoRA ($r = 5$) | 0.269 | 3 M |
| LoRA ($r = 3$) | 0.274 | 1.8 M |
| Ours ($\tau = 0.02$) | 0.2635 | 1.8 M |
| Ours ($\tau = 0.1$) | 0.272 | 1.5 M |

## 5 Discussion and limitations

This work leverages the geometry of the Tucker tensor factorization manifolds to construct a robust and efficient training algorithm for neural networks in compressed Tucker format. The proposed method has a theoretical backbone of approximation error bounds to the full model and guarantees of loss descent and convergence to stationary points in expectation. The method is superior to standard factorization approaches with alternating or simultaneous gradient descent, as demonstrated in the compression-to-accuracy ratio for various benchmarks and models. The method provides a significant reduction of hyperparameters to a single parameter $\tau$. This parameter has a clear interpretation as compression rate per layer depending on the layer's importance on the overall network training dynamics. We, however, note that further strategies to pick an adequate $\tau$ are possible and remain to be investigated. Further, we note that an efficient implementation on GPUs requires an efficient tensor rounding algorithm in Algorithm 1. Finally, the proposed method assumes well-performing low-rank Tucker sub-nets exist in the reference network. While we observe this empirically, further investigations are required to provide theoretical evidence supporting this assumption, similar to the case of fully-connected linear layers, see [3, 6].

## Acknowledgments and Disclosure of Funding

This manuscript has been authored by UT-Battelle, LLC under Contract No. DE-AC05-00OR22725 with the U.S. Department of Energy. The United States Government retains and the publisher,by accepting the article for publication, acknowledges that the United States Government retains a non-exclusive, paid-up, irrevocable, world-wide license to publish or reproduce the published form of this manuscript, or allow others to do so, for United States Government purposes. The Department of Energy will provide public access to these results of federally sponsored research in accordance with the DOE Public Access Plan.

The work of E. Zangrando was funded by the MUR-PNRR project "Low-parametric machine learning". Francesco Tudisco is partially funded by the project FIN4GEO within the European Union's Next Generation EU framework, Mission 4, Component 2, CUP P2022BNB97.

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

# A Impact statement

This paper presents work whose main goal is to reduce the training costs of tensor-based architecture, while maintaining mathematically sound guarantees of performance in terms of convergence and approximation. As in the majority of deep learning research, there are potential societal consequences of our work, none of which we feel must be specifically highlighted here. A positive societal impact is the reduces carbon emissions by more efficient neural networkt training and inference. Regarding ethical aspects, we feel nothing has to be added.

# B Algorithm for tensor taping based on Theorem 2.1

For the sake of completeness, we provide the algorithmic description of the method to generate gradients of the Tucker factors as obtained by Theorem 2.1.

The training algorithm for tensor-valued neural network layers in Tucker format is presented in Algorithm 2. Through the lens of computational efficiency, the main difference to Algorithm 1 is the following:

Each time we back-propagate through a convolutional layer $W = C \times_{i=1}^{d} U_i$, we form the new variable $K_i = U_i S_i$ as in Theorem 2.1, we integrate the ODE in (5) from $K_i(0) = K_i$ to $K_i(\lambda)$, $\lambda > 0$, and then update the factors $U_i$ by forming an orthonormal basis of the range of $K_i(\lambda)$. This strategy directly follows from Theorem 2.1.

Similar to Algorithm 1, we implement the orthonormalization step via the QR factorization while we perform the integration of the gradient flow via stochastic gradient descent with momentum and learning rate $\lambda$, which coincides with a stable two-step linear multistep integration method [69]. Once all the factors $U_i$ are updated, we back-propagate the core term by integrating the equation for $C$ in (5), using the same approach.

Similar to Algorithm 1, the Tucker rank of the new kernel can be adaptively learned with a key basis-augmentation trick. The implementation for Algorithm 2 works as follows: Each time we backpropagate $K_i \in \mathbb{R}^{n_i \times r_i}$, we form an augmented basis $\widetilde{K}_i$ by appending the previous $U_i$ to the new $K_i(\lambda)$, $\widetilde{K}_i = [K_i | U_i]$. We compute an orthonomal basis $U_i^{\text{new}} \in \mathbb{R}^{n_i \times 2r_i}$ for $\widetilde{K}_i$ and we form the augmented $2r_1 \times \cdots \times 2r_d$ core $\widetilde{C} = C \times_{i=1}^{d} (U_i^{\text{new}})^{\top} U_i$. We then backpropagate the core $C$ integrating (5) starting from $C(0) = \widetilde{C}$. Finally, we perform a rank adjustment step by computing the best Tucker approximation of $\widetilde{C}$ to a relative tolerance $\tau > 0$. This step corresponds to solving the following optimization (rounding) task:

$$\text{Find } \widehat{C} \in \mathcal{M}_{\leq 2\rho} \text{ of smallest rank } \rho' = (r_1', \ldots, r_d') \text{ such that } \quad \|\widetilde{C} - \widehat{C}\| \leq \tau \|\widetilde{C}\|$$

where $\rho = (r_1, \ldots, r_d)$ and $\mathcal{M}_{\leq 2\rho}$ denotes the set of tensors with component-wise Tucker rank lower than $2\rho$. In practice, this is done by unfolding the tensor along each mode and computing a truncated SVD of the resulting matrix. The tensor $\widehat{C} \in \mathcal{M}_{\rho'}$ is then further decomposed in its Tucker decomposition yielding a factorization $\widehat{C} = C' \times_{i=1}^{d} U_i' \in \mathcal{M}_{\rho'}$. The parameter $\tau$ is responsible for the compression rate of the method, as larger values of $\tau$ yield smaller Tucker ranks and thus higher parameter reduction. To conclude, the computed $U_i' \in \mathbb{R}^{2r_i \times r_i'}$ with $r_i' \leq 2r_i$ are then pulled back to the initial dimension of the filter by setting $U_i = U_i^{\text{new}} U_i' \in \mathbb{R}^{n_i \times r_i'}$, and the new core tensor $C$ is then assigned $C'$.

This implementation shares the robust error bound of Algorithm 1. However it comes at an increased computational cost due to $d + 1$ necessary evaluations of the network and gradient tape, where the first $d$ are due to the basis updates $K_i$ and the last for the coefficient update $S$.

# C Additional experiments

## C.1 Additional experiments for ResNet18 on Tiny-Imagenet

Table 3 displays the compression to test accuracy results for ResNet18 on Tiny ImageNet as a supplement to the results in section 4.1.

**Algorithm 2:** TDLRT: Standard Dynamical Low-Rank Training of convolutions in Tucker format.

**Input :** Initial low-rank factors $C \sim r_1 \times \cdots \times r_d$; $U_i \sim n_i \times r_i$;
  adaptive: Boolean flag that decides whether or not to dynamically update the ranks;
  $\tau$: singular value threshold for the adaptive procedure.

1 **for** *each mode $i$* **do**
2     $Q_i S_i^\top \leftarrow$ QR decomposition of $\mathrm{Mat}_i(C)^\top$
3     $K_i \leftarrow U_i S_i$
4     $K_i \leftarrow$ descent step; direction $\nabla_{K_i}\mathcal{L}(\mathrm{Ten}_i(Q_i^\top) \times_{j \neq i} U_j \times_i K_i)$; starting point $K_i$
5     **if** adaptive **then**                             /* Basis augmentation step */
6        |   $K_i \leftarrow [K_i \,|\, U_i]$
7     $U_i^{\mathrm{new}} \leftarrow$ orthonormal basis for the range of $K_i$
8 $\widetilde{C} \leftarrow C \times_{i=1}^d (U_i^{\mathrm{new}})^\top U_i$
9 $C \leftarrow$ descent step; direction $\nabla_C \mathcal{L}\big(\widetilde{C} \times_{i=1}^d U_i^{\mathrm{new}}\big)$; starting point $\widetilde{C}$

10 **if** adaptive **then**                                        /* Rank adjustment step */
11     $(C, U_1, \ldots, U_d) \leftarrow$ Tucker decomposition of $C$ up to relative error $\tau$
12     $U_i \leftarrow U_i^{\mathrm{new}} U_i$, for $i = 1, \ldots, d$
13 **else**
14     $U_i \leftarrow U_i^{\mathrm{new}}$, for $i = 1, \ldots, d$

Table 3: Tiny-imagenet benchmark with ResNet18. TDLRT outperforms standard Tucker factorization in terms of the compression-to-accuracy ratio.

|  | test acc [%] | c.r. [%] |
|---|---|---|
| Baseline | 51.1 | 0.0 |
| TDLRT, $\tau = 0.02$ | 50.90 | 83.95 |
| TDLRT, $\tau = 0.06$ | 49.32 | **92.12** |
| Tucker-factorized | 49.66 | 83.66 |

### C.2 Additional experiments VGG16 on Cifar10

The rank evolution over the optimization steps of VGG16 on Cifar10 are displayed in Figure 3. The color gradients indicate the position of the respective tensor basis in the network, where lighter green denotes bases near the input and darker green denotes bases near the output layer of VGG16. Higher singular value cutoff tolerance $\tau$ results in faster rank decay; however, across different choices of tau, a monotonous decrease in the ranks to a steady state is observable.

The proposed method Algorithm 1 allows us to choose any Tucker ranks at initialization In Figure 3, we initialize the network layers with full rank. The decay of the layers' ranks over time is typical for Alg1 and, indeed observed for other architectures as well. This is aligned with theoretical [5] and empirical [18] findings stating that neural networks trained with SGD exhibit a low-rank structure. The results of Fig 3 indicate that Alg. 1 can identify this low-rank manifold during training.

### C.3 Additional experiments for AlexNet on Cifar 10

Tab.4 contains the Tucker ranks of Alexnet compressed with TDLRT as a supplement to the results presented in section4.1. All the test cases were run with the rank adaptive version of the integrator.

For Cifar10, a standard random crop and random flip are used for data augmentation at training time. All methods are trained for 70 epochs using a batch size of 128. All methods are trained using the SGD optimizer, with a starting learning rate of $5 \times 10^{-2}$ with a scheduler that reduces it by a factor of 10 whenever validation loss reaches a plateau. Polyak momentum was 0.1 for all layers but batch normalizations, which was set to 0.9.

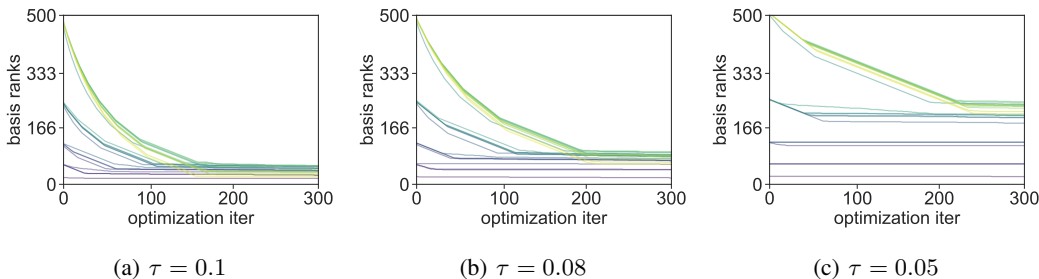

(a) $\tau = 0.1$       (b) $\tau = 0.08$       (c) $\tau = 0.05$

Figure 3: Rank evolution of the Tucker bases over optimization steps of all rank adaptive layers of VGG16 for Cifar10 using Algorithm 1. The lighter color indicates ranks of bases of deeper layers of the network. A higher singular value cutoff threshold $\tau$ results in faster rank decay and smaller steady-state ranks, leading to a potentially higher compression rate.

Table 4: Reproduction of the results of Alexnet on Cifar10. The ranks reported refer to the Tucker ranks of each convolutional layer.

|  | test acc.[%] | layers' ranks | test compression rate [%] |
|---|---|---|---|
| Baseline | 79.63 | $[64, 3, 3, 3]$ $[192, 64, 3, 3]$ $[384, 192, 3, 3]$ $[256, 384, 3, 3]$ $[256, 256, 3, 3]$ | 0.0 |
| TDLRT $\tau = 0.6$ | 76.26 | $[25, 3, 3, 3]$ $[76, 25, 3, 3]$ $[153, 76, 3, 3]$ $[102, 153, 3, 3]$ $[102, 102, 3, 3]$ | 74 |
| TDLRT $\tau = 0.7$ | 73.08 | $[19, 3, 3, 3]$ $[57, 19, 3, 3]$ $[115, 57, 3, 3]$ $[76, 115, 3, 3]$ $[76, 76, 3, 3]$ | 83.5 |

For more complicated datasets and architectures like Cifar10 on VGG16 and Alexnet, results in Tab.1 and Fig.1 show that our proposal consistently gets better results than all the methods under comparison at parity of compression.

## D    Proof of Theorem 2.1

*Theorem.* Let $W = C \times_{i=1}^{d} U_i \in \mathcal{M}_\rho$ be such that (3) holds. Let $\mathrm{Mat}_i(C)^\top = Q_i S_i^\top$ be the QR decomposition of $\mathrm{Mat}_i(C)^\top$ and let $K_i = U_i S_i$. Then,

$$\dot{K}_i = -\nabla_{K_i} \mathcal{L}\big( \mathrm{Ten}_i(Q_i^\top) \times_{j \neq i} U_j \times_i K_i \big) \qquad \text{and} \qquad \dot{C} = -\nabla_C \mathcal{L}(C \times_{j=1}^{d} U_j) \qquad (8)$$

where $\mathrm{Ten}_i$ denotes "tensorization along mode $i$", i.e. the inverse reshaping operation of $\mathrm{Mat}_i$.

*Proof.* The proof follows the path first suggested in [52, §4], i.e., the quantity $V_i$ defined next is frozen in time: $\dot{V}_i = 0$. A detailed derivation of the matrix-tensor equations for $K_i$ and $C$ is beyond the scope of the present work and it is provided in [40, 53, 54, 12, 11].

We begin recalling the evolution equations for the factors of the projected gradient flow (3):

$$\begin{cases} \dot{U}_i &= -(I - U_i U_i^\top) \mathrm{Mat}_i\big(\nabla_W \mathcal{L}(W) \times_{j \neq i} U_j^\top\big) \mathrm{Mat}_i(C)^\dagger, \quad i = 1, \dots, d \\ \dot{C} &= -\nabla_W \mathcal{L}(W) \times_{j=1} U_j^\top. \end{cases}$$

where † denotes the pseudoinverse. We assume that the tensor $W$ admits a differentiable Tucker tensor representation, i.e., $W(t) = C(t) \times_{i=1}^{d} U_i(t) \in \mathcal{M}_\rho$ for $t \in [0, \lambda]$ satisfying the associated gradient-flow tensor differential equation

$$\dot{W}(t) = -\nabla_W \mathcal{L}(W(t)).$$

For sake of brevity, the parameter is going to be omitted. Next, we perform a QR-factorization

$$\mathrm{Mat}_i(C)^\top = Q_i S_i^\top.$$

Thus, we observe that the matrix $\mathrm{Mat}_i(W)$ admits an SVD-like decomposition as follows

$$\mathrm{Mat}_i(W) = U_i S_i V_i^\top \quad \text{with} \quad V_i^\top := Q_i^\top \bigotimes_{j \neq i} U_j^\top,$$

where the matrix $Q_i$ possess othornormal columns, i.e., $Q_i^\top Q_i = I$. We introduce the quantity

$$K_i := U_i S_i \quad \text{where} \quad S_i = \mathrm{Mat}_i(C)Q_i.$$

By construction, we observe that

$$S_i = U_i^\top \mathrm{Mat}_i(W)V_i.$$

The tiny matrix $S_i$ satisfies the following differential equation

$$\begin{aligned}
\dot{S}_i &= \dot{U}_i^\top \mathrm{Mat}_i(W)V_i + U_i^\top \mathrm{Mat}_i(\dot{W})V_i + U_i^\top \mathrm{Mat}_i(W)\dot{V}_i \\
&= \underbrace{\left( \dot{U}_i^\top U_i \right)}_{=0} S_i V_i^\top V_i + U_i^\top \mathrm{Mat}_i(\dot{W})V_i + U_i^\top \mathrm{Mat}_i(W) \underbrace{\dot{V}_i}_{=0} \\
&= -U_i^\top \mathrm{Mat}_i(\nabla_W \mathcal{L})V_i.
\end{aligned}$$

The first null identity follows from the gauge condition on $U_i$. The second null identity follows by the initial assumption $\dot{V}_i = 0$. Therefore, the matrix $K_i$ satisfies the differential equation

$$\begin{aligned}
\dot{K}_i &= (\dot{U_i S_i}) \\
&= \dot{U}_i S_i + U_i \dot{S}_i \\
&= -(I - U_i U_i^\top) \mathrm{Mat}_i(\nabla_W \mathcal{L} \times_{j \neq i} U_j^\top) \mathrm{Mat}_i(C)^\dagger S_i - U_i U_i^\top \mathrm{Mat}_i(\nabla_W \mathcal{L})V_i \\
&= (U_i U_i^\top - I) \mathrm{Mat}_i(\nabla_W \mathcal{L} \times_{j \neq i} U_j^\top)Q_i - U_i U_i^\top \mathrm{Mat}_i(\nabla_W \mathcal{L})V_i \qquad (9) \\
&= (U_i U_i^\top - I) \mathrm{Mat}_i(\nabla_W \mathcal{L}) \cdot (\bigotimes_{j \neq i} U_j)Q_i - U_i U_i^\top \mathrm{Mat}_i(\nabla_W \mathcal{L})V_i \\
&= (U_i U_i^\top - I) \mathrm{Mat}_i(\nabla_W \mathcal{L})V_i - U_i U_i^\top \mathrm{Mat}_i(\nabla_W \mathcal{L})V_i \\
&= -\mathrm{Mat}_i(\nabla_W \mathcal{L})V_i.
\end{aligned}$$

To conclude, we set $Z = K_i V_i^\top$ and let $W = \mathrm{Ten}_i(K_i V_i^\top) = \mathrm{Ten}_i(Z)$. We obtain

$$\nabla_{K_i} \mathcal{L}(W) = \nabla_Z \mathcal{L}(\mathrm{Ten}_i(Z))\nabla_Z K_i = \nabla_Z \mathcal{L}(\mathrm{Ten}_i(Z))V_i.$$

where we remind that $V_i^\top V_i = I$. Hence

$$\mathrm{Mat}_i(\nabla_W \mathcal{L}(W))V_i = \nabla_Z \mathcal{L}(\mathrm{Ten}_i(Z))V_i = \nabla_{K_i} \mathcal{L}(W). \qquad (10)$$

The first $K_i$-differential equation is then obtained combining (11) and (10)

$$\dot{K}_i = -\mathrm{Mat}_i(\nabla_W \mathcal{L})V_i = -\nabla_{K_i} \mathcal{L}\left(\mathrm{Ten}_i(K_i V_i^\top)\right). \qquad (11)$$

The right-hand side can be further reduced using standard tensorization formulas [41]

$$\mathrm{Ten}_i(K_i V_i^\top) = \mathrm{Ten}_i(Q_i^\top) \underset{j \neq i}{\times} U_j \times_i K_i.$$

The second differential equations follows by observing that

$$\nabla_W \mathcal{L} = \nabla_C \mathcal{L} \times_i U_i + \sum_j \mathcal{L} \times_j \dot{U}_j \times_{i \neq j} U_i.$$

The tensor $C$ satisfies the differential equation

$$
\begin{aligned}
\dot{C} &= -\nabla_W \mathcal{L}(W) \times_i U_i^\top \\
&= -\nabla_C \mathcal{L}(W) \times_i U_i \times_i U_i^\top \\
&= -\nabla_C \mathcal{L}(W) \times_i \underbrace{U_i^\top U_i}_{=I} \\
&= -\nabla_C \mathcal{L}(C \times_i U_i) \,.
\end{aligned}
$$

where the extra terms disappear due to the imposed gauge conditions $U_i^\top \dot{U}_i = 0$. $\qquad\square$

## E    Proof of Corollary 2.2

**Corollary E.1.** *Let* $W = C \times_{i=1}^d U_i \in \mathcal{M}_\rho$ *be such that (6) holds. Let* $\mathrm{Mat}_i(C)^\top = Q_i S_i^\top$ *be the QR decomposition of* $\mathrm{Mat}_i(C)^\top$ *and let* $K_i = U_i S_i$. *Then,*

$$
\mathrm{span}\left(\left[U_i, \dot{K}_i\right]\right) = \mathrm{span}\left(\left[U_i, \nabla_{U_i} \mathcal{L}(W)\right]\right). \tag{12}
$$

*Proof.* From Eq. (9) of the proof in Theorem 2.1 it is apparent, that

$$
\dot{K}_i = \dot{U}_i S_i + U_i \dot{S}_i = -\mathrm{Mat}_i(\nabla_W \mathcal{L}(Ten_i(W))) V_i. \tag{13}
$$

Moreover, we observe that using the chain rule,

$$
\nabla_{U_i} \mathcal{L} = \mathrm{Mat}_i(\nabla_W \mathcal{L}) \nabla_{U_i}(U_i S_i V_i^\top) = \mathrm{Mat}_i(\nabla_W \mathcal{L}) V_i S_i^\top
$$

Then, with (13) we have

$$
\nabla_{U_i} \mathcal{L} = \mathrm{Mat}_i(\nabla_W \mathcal{L}) V_i S_i^\top = -\dot{K}_i S_i^\top \,.
$$

Using full-rankness of $S_i$ concludes the proof. $\qquad\square$

## F    Proofs of descent and approximation theorems

*Theorem.* Let $W(\lambda) = C \times_{j=1}^d U_j$ be the Tucker low-rank tensor obtained after one training iteration using Algorithm 2 and let $W(0)$ be the previous point. Then, for a small enough learning rate $\lambda$, it holds $\mathcal{L}_W(W(\lambda)) \leq \mathcal{L}_W(W(0)) - \alpha\lambda + \beta\tau$, where $\alpha, \beta > 0$ are constants independent of $\lambda$ and $\tau$, and where $\mathcal{L}_W$ denotes the loss as a function of only $W$.

*Proof.* Let $\widehat{W}(t) = \widehat{C}(t) \times_i \widehat{U}_i^1$. Here, $\widehat{W}(t)$ and $\widehat{C}(t)$ denote the augmented solutions for $t \in [0, \lambda]$ arising from the intermediate steps of the TDLRT Algorithm 2. We observe that

$$
\begin{aligned}
\frac{d}{dt}\mathcal{L}(\widehat{W}(t)) &= \langle \nabla \mathcal{L}(\widehat{W}(t)), \dot{\widehat{W}}(t) \rangle \\
&= \langle \nabla \mathcal{L}(\widehat{W}(t)), \dot{\widehat{C}}(t) \times_i \widehat{U}_i^1 \rangle \\
&= \langle \nabla \mathcal{L}(\widehat{W}(t)) \times_i \widehat{U}_i^{1,\top}, \dot{\widehat{C}}(t) \rangle \\
&= \langle \nabla \mathcal{L}(\widehat{W}(t)) \times_i \widehat{U}_i^{1,\top}, -\nabla \mathcal{L}(\widehat{W}(t)) \times_i \widehat{U}_i^{1,\top} \rangle = -\|\nabla \mathcal{L}(\widehat{W}(t)) \times_i \widehat{U}_i^{1,\top}\|^2 \,.
\end{aligned}
$$

If we define $\alpha := \min_{0 \leq \tau \leq 1} \|\nabla \mathcal{L}(\widehat{W}(\tau\lambda)) \times_i \widehat{U}_i^{1,\top}\|^2$, it follows that for $t \in [0, \lambda]$

$$
\frac{d}{dt}\mathcal{L}(\widehat{W}(t)) \leq -\alpha \,. \tag{14}
$$

Integrating (14) from $t = 0$ until $t = \lambda$, we obtain

$$
\mathcal{L}(\widehat{W}(\lambda)) \leq \mathcal{L}(\widehat{W}(0)) - \alpha\lambda.
$$

Because the augmented subspaces $\widehat{U}_i$ contain by construction the range and co-range of the initial value, we have that $\widehat{W}(0) = W(0)$. Furthermore, the truncation is such that $\|W(\lambda) - \widehat{W}(\lambda)\| \leq \tau$. Therefore

$$
\mathcal{L}(W(\lambda)) \leq \mathcal{L}(\widehat{W}(\lambda)) + \beta\tau
$$

where $\beta = \max_{0 \leq \tau \leq 1} \|\nabla \mathcal{L}(\tau W(\lambda) + (1 - \tau)\widehat{W}(\lambda))\|$. $\qquad\square$

**Lemma F.1.** *The following estimate holds*

$$\|W(\lambda) - W(\lambda) \times_j U_j(\lambda)U_j(\lambda)^\top\| \le \Theta = \mathcal{O}(h(h+\epsilon)),$$

*where the hidden constants depend only on $L_1$ and $L_2$.*

*Proof.* It has been shown in [68, Appendix] that there exists a constant $\theta \propto \mathcal{O}(h(h+\epsilon))$ such that

$$\|U_j U_j^\top \operatorname{Mat}_j(W(\lambda)) - \operatorname{Mat}_j(W(\lambda))\| \le \theta \qquad \forall j = 1, \ldots, d,$$

where the value $\theta$ has been shown to depend only on the constants $L_1$, $L_2$ and $\lambda$. The proof of the Lemma follows a recursive constructive argument

$$\|W(\lambda) - W(\lambda) \times_j^d U_j(\lambda)U_j(\lambda)^\top\|$$
$$\le \|W(\lambda) - W(\lambda) \times_j^{d-1} U_j(\lambda)U_j(\lambda)^\top\| + \|W(\lambda) \times_j^{d-1} U_j(\lambda)U_j(\lambda)^\top - W(\lambda) \times_j^d U_j(\lambda)U_j(\lambda)^\top\|$$
$$\le \|W(\lambda) - W(\lambda) \times_j^{d-1} U_j(\lambda)U_j(\lambda)^\top\| + \|(W(\lambda) \times_d U_d(\lambda)U_d(\lambda)^\top - W(\lambda)) \times_j^{d-1} U_j(\lambda)U_j(\lambda)^\top\|$$
$$\le \|W(\lambda) - W(\lambda) \times_j^{d-1} U_j(\lambda)U_j(\lambda)^\top\| + \|W(\lambda) \times_d U_d(\lambda)U_d(\lambda)^\top - W(\lambda)\|$$
$$\le \|W(\lambda) - W(\lambda) \times_j^{d-1} U_j(\lambda)U_j(\lambda)^\top\| + \theta.$$

The conclusion is obtained iterating the provided argument. □

**Theorem F.2.** *For an integer $k$, let $t = k\lambda$, and let $W(t)$ be the full convolutional kernel, solution of (2) at time $t$. Let $C(t), \{U_i(t)\}_i$ be the Tucker core and factors computed after $k$ training steps with Algorithm 2, where the one-step integration from $0$ to $\lambda$ is done exactly. Finally, assume that for any $Y$ in a neighborhood of $W(t)$, the gradient flow $-\nabla \mathcal{L}_W(Y)$ is "$\varepsilon$-close" to $\mathcal{M}_\rho$. Then,*

$$\|W(t) - C(t) \times_{j=1}^d U_j(t)\| \le c_1\varepsilon + c_2\lambda + c_3\tau/\lambda$$

*where the constants $c_1$, $c_2$ and $c_3$ depend only on $L_1$ and $L_2$.*

*Proof.* First, we provide a bound for the local error, i.e., the error obtained after one training epoch. If we apply Lemma F.1, we obtain that

$$\|W(\lambda) - C(\lambda) \times_{j=1}^d U_j(\lambda)\|$$
$$\le \|W(\lambda) - W(\lambda) \times_{j=1}^d U_j(\lambda)U_j(\lambda)^\top\| + \|W(\lambda) \times_{j=1}^d U_j(\lambda)U_j(\lambda)^\top - C(\lambda) \times_{j=1}^d U_j(\lambda)\|$$
$$\le \Theta + \|(W(\lambda) \times_{j=1}^d U_j(\lambda)^\top - C(\lambda)) \times_{j=1}^d U_j(\lambda)\|$$
$$\le \Theta + \|W(\lambda) \times_{j=1}^d U_j(\lambda)^\top - C(\lambda)\|.$$

It suffices to study the latter term. For $t \in [0, \lambda]$, we define the quantity

$$\widetilde{C}(t) := W(t) \times_{j=1}^d U_j(\lambda)^\top$$

It satisfies the differential initial value problem

$$\dot{\widetilde{C}} = -\nabla_W \mathcal{L}(W) \times_{j=1}^d U_j(\lambda)^\top, \quad C(0) = W(0) \times_{j=1}^d U_j(\lambda)^\top.$$

The term $W(t)$ can be written as a perturbation of $\widetilde{C}$

$$W(t) = \widetilde{C} \times_{j=1}^d U_j(\lambda) + R(t),$$

where

$$R(t) = W(t) - W(t) \times_{j=1}^d U_j(\lambda)U_j(\lambda)^\top.$$

Then, we observe that

$$\|W(t) - W(\lambda)\| \le \int_0^\lambda \|\dot{W}(s)\| ds = \int_0^\lambda \|-\nabla_W \mathcal{L}(W(s))\| ds \le C_1\lambda.$$

The remainder can be estimated as follows

$$\|R(t)\| \le \|R(t) - R(\lambda)\| + \|R(\lambda)\| \le 2L_1\lambda + 2\Theta.$$

Furthermore, the full gradient can be re-written as

$$\nabla_W \mathcal{L}(W(t)) = \nabla_W \mathcal{L}(\widetilde{C}(t) \times_{j=1}^d U_j(\lambda) + R(t)) = \nabla_W \mathcal{L}(\widetilde{C}(t) \times_{j=1}^d U_j(\lambda)) + D(t),$$

where the defect $D(t)$ is defined as

$$D(t) := \nabla_W \mathcal{L}(\widetilde{C}(t) \times_{j=1}^d U_j(\lambda) + R(t)) - \nabla_W \mathcal{L}(\widetilde{C}(t) \times_{j=1}^d U_j(\lambda)).$$

Because of the Lipschitiz assumption, we have that

$$\|D(t)\| \le L_2 \|R(t)\| \le 2L_2(L_1\lambda + \Theta).$$

Next, we compare the two differential equations

$$\begin{cases} \dot{\widetilde{C}}(t) = -\nabla_W \mathcal{L}(\widetilde{C}(t) \times_{j=1}^d U_j) \times_{j=1}^d U_j^\top + D(t), \\ \dot{C}(t) = -\nabla_W \mathcal{L}(C(t) \times_{j=1}^d U_j) \times_{j=1}^d U_j^\top, \end{cases}$$

where $C(0) = \widetilde{C}(0)$, by construction. The solution $C(\lambda)$ of the second tensor-differential equation coincides with the solution of the last training step of the TuckerDLRT algorithm. The first differential equation has been constructed such that its solution is $\widetilde{C}(\lambda) = W(\lambda) \times_j U_j^\top$. Therefore, the study of the local error follows by a direct application of the Gronwall inequality

$$\|C(\lambda) - \widetilde{C}(\lambda)\| \le \exp(C_2\lambda) 2L_2(L_1\lambda + \Theta)\lambda.$$

To conclude, the global error in the training epochs follows by using the Lipschitz continuity of the gradient flow: We move from the local error in time to the global error in time by a standard ODEs argument of Lady Windermere's fan [82, §II.3]. □

# G   Proof of stochastic convergence

In this section, we provide the details of the proof of convergence to stationary points in the stochastic setting, Theorem 3.2. The proof extends the approach of [31] to the tensor case and relaxes some of the assumptions there made on the matrix case, while following the same overall structure.

**Theorem G.1** (Convergence). *Let $\widetilde{W}(t)$ be the weight tensor after $t \in \mathbb{N}$ iterations of Algorithm 1 before the rank truncation step, and $W(t)$ as the one obtained after the rank truncation. Assuming that*

- *Algorithm 1 is implemented using SGD as the descent method.*

- *The loss function is assumed to be positive, locally bounded, and differentiable with a Lipschitz gradient.*

- *The learning rate sequence $\lambda_t$ satisfies the Robbins-Monro conditions, i.e.*

$$\sum_t \lambda_t = +\infty \qquad \sum_t \lambda_t^2 < +\infty.$$

- *The spectral distribution stabilizes fast enough over time, i.e.*

$$\sum_{t \ge 0} \mathbb{E}\left[\|\widetilde{W}(t) - W(t)\|\right] < +\infty \tag{15}$$

- *The projected stochastic gradient has a controlled drift, namely*

$$\mathbb{E}\left[\|\nabla \mathcal{L}(W(t-1)) \times_j P_{\widetilde{U}_j(t)}\|^2 \,|\, t-1\right] \le \mu + \nu \|\nabla \mathcal{L}(W(t-1)) \times_j P_{U_j(t-1)}\|^2 \quad \textit{for some } \mu, \nu \ge 0, \tag{16}$$

*where $P_U = UU^T$ is the orthogonal projection onto the range of $U$, and $\mathbb{E}[\cdot|t] = \mathbb{E}[\cdot|W(t), \{U_i(t)\}_{i=1}^d]$ denotes the conditional expectation. Then the following convergence condition holds*

$$\liminf_{t \to \infty} \mathbb{E}\left[\|\nabla \mathcal{L}(W(t-1)) \times_j P_{U_j(t-1)}\|^2\right] = 0$$

The convergence proof of Theorem 3.2 is based on the following technical lemmas.

**Lemma G.2.** *Let $\mathcal{L}$ be a differentiable loss function, and assume that its gradient $\nabla\mathcal{L}(W)$ is one-sided Lipschitz continuous with constant $L_2$. Then, for any $W$ and $W'$, the following inequality holds*

$$\mathcal{L}(W) \leq \mathcal{L}(W') + \langle \nabla\mathcal{L}(W'), W - W' \rangle + \frac{L_2}{2}\|W - W'\|^2$$

*Proof.* By using Cauchy-Schwartz inequality, we have that

$$\mathcal{L}(W) = \mathcal{L}(W') + \int_0^1 \frac{d}{dt}\mathcal{L}\Big(W' + t(W - W')\Big)\, dt$$

$$= \mathcal{L}(W') + \langle \nabla\mathcal{L}(W'), W - W' \rangle - \int_0^1 \langle \nabla\mathcal{L}(W') - \nabla\mathcal{L}\big((W' + t(W - W')\big), W - W' \rangle\, dt$$

$$\leq \mathcal{L}(W') + \langle \nabla\mathcal{L}(W'), W - W' \rangle + L_2\|W - W'\|^2 \int_0^1 t\, dt$$

$$= \mathcal{L}(W') + \langle \nabla\mathcal{L}(W'), W - W' \rangle + \frac{L_2}{2}\|W - W'\|^2\,.$$

$\square$

**Lemma G.3.** *Let $\widetilde{U}_{j,1} = [U_{j,1}|U_{j,0}]$ be a given basis set with orthonormal columns, $\mathcal{L}$ denote the loss function computed on the whole dataset, and $\mathcal{L}_B$ denote the loss calculated on a batch $B$. Then, for any $j^* \in \{1, \ldots, d\}$ and $W$, it holds that*

$$\mathbb{E}\Big[ \big\langle \nabla\mathcal{L}(W),\, \nabla\mathcal{L}_B(W) \times_{j \neq j^*} P_{U_{j,0}} \times_{j^*} P_{U_{j^*,1}} \big\rangle \Big] \geq 0$$

*Proof.* We introduce first the function

$$\phi(\widehat{U}) := \Big\langle \nabla\mathcal{L}(W) \times_{j \neq j^*} P_{U_{j,0}} \times_{j^*} P_{\widehat{U}}, \nabla\mathcal{L}_B(W) \times_{j \neq j^*} P_{U_{j,0}} \times_{j^*} P_{\widehat{U}} \Big\rangle\,.$$

Let $m$ be the the infimum on the set $\mathcal{U}$ as defined below

$$m = \inf_{\widehat{U} \in \mathcal{U}} \phi(\widehat{U}) \quad \text{with} \quad \mathcal{U} = \{U \in \mathbb{R}^{n_{j^*} \times r_{j^*}} \mid \operatorname{rank}(U) \leq r_{j^*}\}\,.$$

Because the term $\nabla\mathcal{L}(W)$ can be decomposed into a sum of $\nabla\mathcal{L}_B(W) \times_{j \neq j^*} P_{U_{j^*,0}} \times_{j^*} P_{U_{j,1}}$ and its orthogonal component, we observe that the infimum satisfies

$$m \leq \phi(U_{j^*,1}) = \Big\langle \nabla\mathcal{L}(W), \nabla\mathcal{L}_B(W) \times_{j \neq j^*} P_{U_{j^*,0}} \times_{j^*} P_{U_{j,1}} \Big\rangle\,.$$

Hence, by taking the expectation on both sides, we can conclude that

$$\mathbb{E}\Big[\big\langle \nabla\mathcal{L}(W), \nabla\mathcal{L}_B(W) \times_{j \neq j^*} P_{U_{j,0}} \times_{j^*}^d P_{U_{j,1}} \big\rangle\Big] \geq \inf_{\widehat{U} \in \mathcal{U}} \mathbb{E}\Big[\phi(\widehat{U})\Big] \geq \inf_{\widehat{U} \in \mathcal{U}} \|\nabla\mathcal{L}(W) \times_{j \neq j^*} P_{U_{j,0}} \times_{j^*} P_{U_{j^*,1}}\|^2$$

The conclusion follows by observing that the most right term in the inequality is positive. $\square$

With the technical lemmas established above, we are now in the position to prove the theorem 3.2.

*Proof. (Thereom 3.2)* To simplify the notation, we will denote by $\mathbb{E}_t[\cdot]$ the conditional expectation $\mathbb{E}_t[\cdot] := \mathbb{E}[\cdot \mid W(t-1)]$. We first remind the reader of two properties of the conditional expectation. Specifically, for any deterministic function $\psi$ and any random variable $X$, we have that

$$\mathbb{E}_t\Big[\psi(W(t-1))\Big] = \psi(W(t-1)), \quad \mathbb{E}\Big[\mathbb{E}_t[X]\Big] = \mathbb{E}[X]\,.$$

We will begin by examining an upper bound for the one-step drift of 2. We denote by $\widetilde{W}(t) = \widetilde{C}(t) \times_j \widetilde{U}_j(t)$ the weight tensor before truncation at step $t \in \mathbb{N}$. As per the assumption, the optimization in the $C$-step of 2 is defined using an SGD update. Therefore, we have

$$\widetilde{C}(t) = \Big(C(t-1)\times_j U_j(t-1) - \lambda_t \nabla\mathcal{L}(W(t-1)))\Big) \times_j \widetilde{U}_j(t)^\top = \Big(W(t-1) - \lambda_t \nabla\mathcal{L}(W(t-1))\Big) \times_j \widetilde{U}_j(t)^\top\,.$$

Hence
$$\widetilde{W}(t) = \widetilde{C}(t) \times_j \widetilde{U}_j(t) = \left( W(t-1) - \lambda_t \nabla \mathcal{L}(W(t-1)) \right) \times_j P_{\widetilde{U}_j(t)}.$$

where $P_U = UU^T$ is the orthogonal projector onto the range of arbitrary matrix $U$. Applying G.2 we have that

$$\mathbb{E}_t \Big[ \mathcal{L}(\widetilde{W}(t)) - \mathcal{L}(W(t-1)) \Big]$$
$$\leq -\lambda_t \mathbb{E}_t \Big[ \langle \nabla \mathcal{L}(W(t-1)), \nabla \mathcal{L}_B(W(t-1)) \times_j P_{\widetilde{U}_j(t)} \rangle \Big] + \frac{\lambda_t^2 L_2}{2} \mathbb{E}_t \Big[ \| \nabla \mathcal{L}_B(W(t-1)) \times_j P_{\widetilde{U}_j(t)} \|^2 \Big]$$

We notice that for the augmented basis $\widetilde{U}_j(t) = [U_j(t-1)|U_j(t)]$, it holds $P_{\widetilde{U}_j(t)} = P_{U_j(t-1)} + P_{U_j(t)}$. When we expand $\mathcal{L}_B(W(t-1)) \times_j P_{\widetilde{U}_j(t)}$ using its sum representation and apply Lemma G.3 to the mixed terms, we obtain

$$\mathbb{E}_t \Big[ \mathcal{L}(\widetilde{W}(t)) - \mathcal{L}(W(t-1)) \Big]$$
$$\leq -\lambda_t \mathbb{E}_t \Big[ \langle \nabla \mathcal{L}(W(t-1)), \nabla \mathcal{L}_B(W(t-1)) \times_j P_{U_j(t-1)} \rangle \Big] + \frac{\lambda_t^2 L_2}{2} \mathbb{E}_t \Big[ \| \nabla \mathcal{L}_B(W(t-1)) \times_j P_{\widetilde{U}_j(t)} \|^2 \Big]$$
$$= -\lambda_t \langle \nabla \mathcal{L}(W(t-1)), \nabla \mathcal{L}(W(t-1)) \times_j P_{U_j(t-1)} \rangle + \frac{\lambda_t^2 L_2}{2} \mathbb{E}_t \Big[ \| \nabla \mathcal{L}_B(W(t-1)) \times_j P_{\widetilde{U}_j(t)} \|^2 \Big]$$
$$= -\lambda_t \| \nabla \mathcal{L}(W(t-1)) \times_j P_{U_j(t-1)} \|^2 + \frac{\lambda_t^2 L_2}{2} \mathbb{E}_t \Big[ \| \nabla \mathcal{L}_B(W(t-1)) \times_j P_{\widetilde{U}_j(t)} \|^2 \Big].$$
$$(17)$$

The locally bounded loss computed on the truncated approximation $W(t)$ is bounded via

$$\mathcal{L}(W(t)) \leq \mathcal{L}(\widetilde{W}(t)) + \langle \nabla \mathcal{L}(sW(t) + (1-s)\widetilde{W}(t)), W(t) - \widetilde{W}(t) \rangle \leq \mathcal{L}(\widetilde{W}(t)) + C \| W(t) - \widetilde{W}(t) \|$$
$$(18)$$

By combining equations (17) and (18), we arrive at the following bound

$$\mathbb{E}_t \Big[ \mathcal{L}(W(t)) - \mathcal{L}(W(t-1)) \Big]$$
$$\leq -\lambda_t \| \nabla \mathcal{L}(W(t-1)) \times_j P_{U_j(t-1)} \|^2 + \frac{\lambda_t^2 L_2}{2} \mathbb{E}_t \Big[ \| \nabla \mathcal{L}_B(W(t-1)) \times_j P_{\widetilde{U}_j(t)} \|^2 \Big] + C \mathbb{E}_t \Big[ \| W(t) - \widetilde{W}(t) \| \Big]$$
$$(19)$$

Following assumption (16), we have

$$\mathbb{E}_t \Big[ \mathcal{L}(W(t)) - \mathcal{L}(W(t-1)) \Big]$$
$$\leq -\lambda_t \| \nabla \mathcal{L}(W(t-1)) \times_j P_{U_j(t-1)} \|^2 + \frac{\lambda_t^2 L_2}{2} \Big( \mu + \nu \| \nabla \mathcal{L}(W(t-1)) \times_j P_{U_j(t-1)} \|^2 \Big) + C \mathbb{E}_t \Big[ \| W(t) - \widetilde{W}(t) \| \Big]$$
$$= -\lambda_t (1 - \frac{1}{2} \lambda_t L_2 \nu) \| \nabla \mathcal{L}(W(t-1)) \times_j P_{U_j(t-1)} \|^2 + \frac{\lambda_t^2 L_2 \mu}{2} + C \mathbb{E}_t \Big[ \| W(t) - \widetilde{W}(t) \| \Big]$$
$$\leq -\lambda_t \| \nabla \mathcal{L}(W(t-1)) \times_j P_{U_j(t-1)} \|^2 + \frac{\lambda_t^2 L_2 \mu}{2} + C \mathbb{E}_t \Big[ \| W(t) - \widetilde{W}(t) \| \Big],$$

where we assume that $\lambda_t \leq 2/L_2\nu$. Finally, by taking the expectation on both sides, we obtain

$$\mathbb{E} \Big[ \mathcal{L}(W(t)) - \mathcal{L}(W(t-1)) \Big] \leq -\lambda_t \mathbb{E} \Big[ \| \nabla \mathcal{L}(W(t-1)) \times_j P_{U_j(t-1)} \|^2 \Big] + \frac{\lambda_t^2 L_2 \mu}{2} + C \mathbb{E} \Big[ \| W(t) - \widetilde{W}(t) \| \Big]$$

By summing the last equation over $t = 1, \ldots, T$ we get

$$-\mathcal{L}(W(0)) \leq \mathbb{E} \Big[ \mathcal{L}(W(t)) - \mathcal{L}(W(0)) \Big] \leq$$
$$-\sum_{t=1}^{T} \lambda_t \mathbb{E} \Big[ \| \nabla \mathcal{L}(W(t-1)) \times_j P_{U_j(t-1)} \|^2 \Big] + \frac{L_2 \mu}{2} \sum_{t=1}^{T} \lambda_t^2 + C \sum_{t=1}^{T} \mathbb{E} \Big[ \| W(t) - \widetilde{W}(t) \| \Big]$$

By rearranging the terms of the latter equation and by sending $T \to +\infty$, we finally obtain

$$\sum_{t=1}^{+\infty} \lambda_t \mathbb{E}\left[\|\nabla\mathcal{L}(W(t-1)) \times_j P_{U_j(t-1)}\|^2\right] \leq$$

$$\mathcal{L}(W(0)) + \frac{L_2\mu}{2}\sum_{t=1}^{+\infty}\lambda_t^2 + C\sum_{t=1}^{+\infty}\mathbb{E}\left[\|W(t) - \widetilde{W}(t)\|\right] < +\infty$$

The conclusion follows by the Robbins-Monro conditions, i.e.

$$\liminf_{t\to\infty}\mathbb{E}\left[\|\nabla\mathcal{L}(W(t-1)) \times_j P_{U_j(t-1)}\|^2\right] = 0$$

$\square$

