# OpenReview forum: "Geometry-aware training of factorized layers in tensor Tucker format"
_NeurIPS.cc/2024/Conference — NeurIPS 2024 poster_

### Official Review · Reviewer_Rp73 · 2024-07-07

**Soundness:** 3
**Presentation:** 3
**Contribution:** 3
**Rating:** 6
**Confidence:** 2

**Summary:**

This paper proposes a method for neural network model parameter compression. It parameterizes weight tensors in the format of Tucker decomposition, and trains the factors instead of the origin weight tensors afterwards. The method is able to adaptively modify the tensor rank. Authors also provide detailed theoretical analysis including the computational steps, and convergence, approximation and the gradient descent guarantees of the method. Experiments show good performance of the method in terms of image classification accuracies, model compressive rates, and the high running efficiency.

**Strengths:**

Writing is easy to follow. Theoretical analysis is self-contained and solid.  Experiments are sufficient to demonstrate the claimed good properties of the method.

**Weaknesses:**

I did not observe obvious weakness. However, I found that the studied neural networks seemed to be a bit of out-of-date. As a reviewer, I hope to see applying the method in up-to-date models e.g. transformers, in other fields. I am not sure whether this point would make negative influence on the significance of the work.

**Questions:**

What is a standard Tucker decomposition like? Could you provide more details for the comparison of the proposed method and the vanilla standard Tucker decomposition, especially over the space/time complexity comparison, and their convergency analysis?

What are the compressed parameters in the neural networks? Are the parameters merely the convolutional kernel? Can the method be applied on the weights of linear layers?

What does ``geometry-aware’’ indicates? What kind of geometry is the method aware of? Could you give more explanation?

---

> ### Author Rebuttal · Authors · 2024-08-06
>
> First of all, we would like to thank the reviewer for their feedback.
>
> **W1**: We agree that our paper can be improved by showcasing stronger empirical evaluations for larger models. We have therefore conducted several new experiments on the popular parameter-efficient fine-tuning
> using low-rank adapters (LoRA), as suggested by reviewer vGH1.
> Here, we have tested the proposed training model on larger
> architectures such as DebertaV3 on Glue benchmark and Stable Diffusion
> (v1.4) dreambooth. In DebertaV3, we applied our method to matrix layers
> by interpreting matrices as order-2 tensors. For Stable Diffusion we
> applied our method to tensor layers in the Unet, while standard LoRA
> convolution works essentially by reshaping the tensor into a matrix.
> We believe that these results strengthen the empirical evaluation and we hope these could address your concerns.
>
> | GLUE test case| LoRA - rank 8, 1.33M params | Ours - 0.9 M params |
> |-----------------------|---------------|-----------------------------|
> | CoLa (Matt. Corr)     | $0.6759$      | $0.7065$                    |
> | MRPC (acc)            | $0.8971$      | $0.9052$                    |
> | QQP (acc)             | $0.9131$      | $ 0.9215$                   |
> | RTE (acc)             | $0.8535$      | $0.8713$                    |
> | SST2 (acc)            | $0.9484$      | $0.9594$                    |
>
>
>  Stable Diffusion  | loss | \# trainable parameters
> ----------------------|---------------|----------------------------------
>  LoRA ($r = 8$)       | $0.260$       | $5$ M
>  LoRA ($r = 5$)       | $0.269$       | $3$ M
>  LoRA ($r = 3$)       | $0.274$       | $1.8$ M
>  ours ($\tau = 0.02$) | $0.2635$    | $1.8$ M
>  ours ($\tau = 0.1$)  | $0.272$       | $1.5$ M
>
> **Q1**. The standard Tucker decomposition is the choice of the parameterization $W(i_1,\dots,i_d) = \sum_{j_1,\dots,j_d} C(j_1,\dots,j_d) U^{(d)}(i_d,j_d)$ to represent the d-mode tensor $W$. This choice is one possible extension of the classical singular value decomposition for tensors. For what concerns the vanilla Tucker decomposition and its training, we compare with the prototype method used often in training low-rank decompositions (such as LoRA), for which the update is simply given by a step of stochastic gradient descent on each factor of the decomposition, i.e. $C(t+1) = C(t)- \lambda_t \nabla_C \mathcal L(t),\, U^{(i)}(t+1) = U^{(i)}(t)-\lambda_t \nabla_{U^{(i)}}\mathcal L(t)$.
> Apart from the instability with respect to small singular values of vanilla Tucker, we would like to underline that this method introduces additional invariances in the parameter space (since orthonormality of the basis $U^{(i)}$ is not preserved over time). In terms of space complexity, Tucker vanilla and our decomposition are the same. The time complexity of one optimization iteration for the vanilla Tucker is of order $O(b(\prod_{i=1}^d r_i+ \sum_{i=1}^d n_i r_i))$, where $n_i$ are the dimensions of the tensor weight, $r_i$ are the Tucker ranks and $b$ is the batch size. The training of vanilla methods is essentially SGD, so the convergence properties of it are the same of stochastic gradient descent.
>
> **Q2** This is a good question and we will add a small discussion of this point in the manuscript. The presented theory and the algorithm is developed in general for Tucker tensors with $d$ modes, that for $d = 2$ covers also the matrix case. In particular then it is possible to apply the method both to matrices and tensors with more than two modes. In the matrix case, the Tucker decomposition would degenerate into a singular value like decomposition of the weight matrix.
>
> **Q3** The choice of the name "geometry-aware" refers to the fact that the proposed method is "aware" of the geometric structure underlying the problem. In section 2.1 we formulate the training of tensor-weighted neural networks as a gradient flow $\dot W = -\nabla_W \mathcal L(W)$. When we want to superimpose a low-rank structure, vanilla methods follow a gradient flow $\dot C = -\nabla_C \mathcal L, \, \dot U^{(i)} = \nabla_{U^{(i)}} \mathcal L$. This last gradient flow lies in the (Riemannian) manifold $\mathcal M_r$ of Tucker tensors of rank $\mathbf{r}$, but it is not aware in some sense of the original problem related to the geometry of the constraint. Our proposed method follows the global dynamics $\dot W = -P(W) \nabla_W \mathcal L(W)$, where $P(W)$ is the **orthogonal** projection on the tangent space $T_W \mathcal M_r$. Notice that the right hand side of this last projected differential equation is a solution of the minimization problem $ \underset{\delta W \in T_W\mathcal M_r}{\arg \min}||\delta W+ \nabla \mathcal{L}(W) ||_{F}^2$, basically by definition of orthogonal projection. This variational principle is essentially telling us that locally we're taking the closest direction to following the original unfactorized dynamics. This property is not satisfied by the vanilla gradient flow system, that does not follow locally the original unfactorized dynamics.
> Moreover, our proposal is aware of certain properties of $\mathcal M_r$, such as high curvature around tensors with small singular values along some mode. As shown in theorems 3.1 and 3.3, the provided theoretical bounds regarding approximation and descent do not depend on the singular values, making the method stable in their presence. This property is also highlighted numerically in figure 2, where we show that vanilla methods seem to suffer from this ill-conditioning.
>
> We will add some details about the questions in the manuscript, and we hope the explaination has clarified any possible doubt. In any case we remain available if any further clarification is needed.

---

> > ### Comment · Reviewer_Rp73 · 2024-08-10
> >
> > Thanks for your response. I would keep my score.

---

> > > ### Author Response · Authors · 2024-08-12
> > >
> > > We thank again the reviewer for the feedback.

---

### Official Review · Reviewer_vGH1 · 2024-07-10

**Soundness:** 3
**Presentation:** 3
**Contribution:** 3
**Rating:** 7
**Confidence:** 5

**Summary:**

This paper extends the dynamic low-rank neural network training (DLRT) method to the rank-adaptive Tucker tensor format (TDLRT). The proposed reparameterization method greatly reduces the computational complexity and numerical instability of the projected gradient descent. Under certain conditions, TDLRT with SGD converges to a stationary point in expectation, where the tensor found by TDLRT provably approximates the full mode. The experimental results show that TDLRT converges faster with less variance and better accuracy-compression rate tradeoff than other factorization-based and pruning-based methods.

**Strengths:**

- The paper is clearly written. Most parts are supported by sufficient technical details.
- The proposed method is an extended version of DLRT. Yet, some unique challenges appear in the Tucker tensor format have addressed with interesting approaches, e.g., the reparameterization method and Corollary 2.2.

**Weaknesses:**

I have some minor concerns on writing and the experimental result

- More detailed information about one-step integration methods could be provided for the readers who are not familiar with the concept.
- Some sections assume a certain degree of background knowledge on DLRT, e.g., gauge conditions in Line 612.
- The evaluation of training time for Tucker Decomposition and TDLRT was conducted on a toy-sized model (LeNet5) and dataset (MNIST). Evaluating training time on a dataset and a model of more practical sizes would provide better understandings of the proposed method.

**Questions:**

- How does the one-step integration work? Does it not add significant computational overhead?
- Are the conditions and assumptions in Theorem 3.2 and 3.3 reflect the actual DNN training? Is there any assumption that does not hold in practice?
- How sensitive is TLDRT to the choice of hyperparameters like $\tau$, learning rate, weight decay or initialization schemes? Do they require extensive hyperparameter search to find the right ones?
- Can TDLRT be used for the low-rank adaptation (LoRA) [1] setting? E.g., fine-tuning the convolution kernels of the U-Nets for the diffusion models by TDLRT with very low tensor ranks.

[1] Hu, Edward J., et al. "Lora: Low-rank adaptation of large language models." arXiv preprint arXiv:2106.09685 (2021).

**Limitations:**

- The utilization of QR decomposition might hinder using the proposed method to train a large model on a large dataset, e.g., diffusion model training.
- The number of parameters is determined after training and indirectly adjusted by a hyperparameter $\tau$. Since a good strategy of choosing $\tau$ is not yet proposed, one might need to train the DNN multiple times to obtain a desired accuracy and model size.

---

> ### Author Rebuttal · Authors · 2024-08-06
>
> First of all, we would like the thank the reviewer for the insightful feedback. Below, we provide our responses to the main points raised in the review.
>
> **Q1+W1** Regarding "How does the one-step integration work? Does it not add significant computational overhead?". With "one-step integration'' we mean a single step of a gradient-based optimization method or, equivalently, any classical time-integration method. In this work, we use stochastic gradient descent (SGD) which in the time integration view is the explicit Euler method. However, the use of other methods is certainly possible. This step does not add any overhead compared to the full-rank training and, in fact, is one of the parts in our method that allows us to have a significant reduction of computational costs and memory footprint. Note that for the full-rank training, the one-step integrate function is called repeatedly using the full gradient, whereas we only need to call this method with the gradient for a low-rank factor. We will add a sentence to our manuscript to clarify what we mean by the one-step integration method.
>
> **W2** Thank you for the suggestion which we believe makes a lot of sense. We will add a short description to our manuscript.
>
> **Q2** Regarding "Are the conditions and assumptions in Theorem 3.2 and 3.3 reflect the actual DNN training?...". Regarding Theorem 3.3, boundedness and Lipschitz continuity of the gradient are very common assumptions one must make to show most analytic results. Both assumptions are reasonable since one would assume the gradient remains bounded during the optimization and the gradient does not completely change when slightly changing the input. The latter can be achieved using most common activation functions, since it requires a condition on the boundedness of the derivatives. The assumption that the gradient flow remains close to the low-rank manifold is observed empirically, but it is extremely difficult to understand analytically. Numerous experiments have shown that neural networks are usually well-represented by low-rank weights, and some analytic investigations exist for heavily simplified architectures. Regarding Theorem 3.2, the Robbins-Monro conditions are very standard assumptions and can be ensured by the user when picking the learning rate. The stabilization of the spectral distribution over time is something that we observe empirically, though it is hard to show anything on an analytic level. Here, we see that after sufficient iterations of TDLRT, the basis does not change, and most of the change is seen in the core tensor. The drift assumption is meant to play the role of the more standard (but more restrictive) assumption of finite variance, which is common when studying convergence of stochastic methods.
>
> **Q3** This is a good question we should have discussed in our manuscript. First, regarding the tolerance parameter for truncating the core tensor, this is, of course, a hyperparameter that needs to be chosen. We commonly use similar values here in different test cases and we never needed an in-depth hyperparameter search to observe good performance. However, it is hard to say if this holds for all architectures and datasets, and most likely, one must adapt here. However, we wish to point out that a single parameter determines individual ranks and compressions for different modes in each tensor in each layer. Here, other approaches, like LoRA, which you mention later, require the user to choose good ranks for each layer, which is certainly a harder task and requires an intensive parameter search if one wants minimal memory overhead. Regarding parameters like learning rate, weight decay etc., we found that our method is relatively robust with respect to their choice, similar to the full baseline. This is aligned with the theoretical findings of e.g. Thm 2.1 and Thm 3.3.
>
> **Q4 + W3** Thank you for this excellent question. Certainly, our approach fits LoRA excellently, and **we have now provided some results on how TDLRT can be used in such a setting** (we refer to the main rebuttal pdf file). The main question is always if these architectures allow for low-rank weights (or in terms of the assumption of Theorem 3.3, if $\varepsilon$ small). If the answer is yes, then our approach should yield good results. Moreover, all the theory presented in the manuscript applies to this case.
>
> **L1** Regarding "The utilization of QR decomposition might hinder using the proposed method to train a large model....'' Please note that the QR decomposition only needs to be computed on a small matrix of dimension $n_i\times r_i$, leading to computational costs of $O(n_i\times r_i^2)$. Thus, if the rank is small, the QR decomposition is usually not a limiting factor (and these small ranks are in fact often the case for LoRA style fine-tuning).
>
> **L2** Regarding "The number of parameters is determined after training and indirectly adjusted by a hyperparameter. Since a good strategy of choosing
> is not yet proposed, one might need to train...". We agree, development of a method that takes a parameter budget and determines $\tau$ to obtain the best model for the given budget is a relevant research question. We have been pointed to  [1] by reviewer TTwz for an approximative Tucker decomposition (of the core tensor) under a parameter budget.
> Here the research question is: Given a parameter budget, how to determine the core shapes of the core tensors of **all** low-rank convolutions in the network to best approximate the full-rank model, if the low-rank dynamics, implicitly given by the data, are representable within this budget.
> This question is as relevant as it is non-trivial, and we would like to refer to future research to properly address it.
>
> [1] Ghadiri, Mehrdad, Matthew Fahrbach, Gang Fu, and Vahab Mirrokni. "Approximately optimal core shapes for tensor decompositions." In International Conference on Machine Learning, pp. 11237-11254. PMLR, 2023.

---

> > ### Comment · Reviewer_vGH1 · 2024-08-09
> >
> > Thank you for your thorough response! I was uncertain about the details of the one-step integration and the practicality of the proposed method, which were all addressed in the rebuttal.
> >
> > In particular, I thank the authors for providing additional experimental results on language and diffusion models. They clarify my doubts about the practicality.
> >
> > Regarding the diffusion model, which layers were updated with LoRA and TDLRT? Although it is common practice to apply LoRA to the attention layers of the U-Net, I suppose TDLRT can also update the convolution layers. I wonder if this was the case, and if so, it could be argued as an additional strength compared to LoRA.
> >
> > Also, I retract Limitation 1 on QR decomposition overhead after rethinking it based on the authors' response. I don't think it will limit the usability of TDLRT as long as the rank is small.
> >
> > Since I already gave 7, I keep my score the same. Instead, I am inclined to raise my confidence.

---

> > > ### Author Response · Authors · 2024-08-12
> > >
> > > We thank again the reviewer for their response and we are glad we were able to clarify all doubts.
> > >
> > > Regarding the U-net, we applied LoRA to the attention and convolutional layers, using the official implementation of LoRA in the Huggingface PEFT package [1]. Specifically, the LoRA implementation for convolutions does not perform a proper tensor decomposition; instead, it corresponds to a low-rank factorization of a flattened version of the convolutional kernel (flattened to a matrix). Our proposed TDLRT has been applied to the same layers as those described above, maintaining their original tensor/matrix structure.
> > >
> > > [1] S. Mangrulkar, S. Gugger, L. Debut, Y. Belkada, S. Paul and B. Bossan, "PEFT: State-of-the-art Parameter-Efficient Fine-Tuning methods", github 2022.
> > >
> > > We thank the reviewer once again for their interest and remain available to clarify any further doubts.

---

### Official Review · Reviewer_jjrQ · 2024-07-12

**Soundness:** 2
**Presentation:** 2
**Contribution:** 3
**Rating:** 5
**Confidence:** 3

**Summary:**

The authors present a novel algorithm for training neural network layers using Tucker tensor decomposition. The approach addresses common issues with layer factorisation, including the need for an initial warm-up phase and sensitivity to parameter initialisation. Th method dynamically updates the ranks during training. The authors provide theoretical guarantees on loss descent, convergence, and approximation to the full model, supported by experimental results showing high compression rates and performance comparable to or better than baseline and alternative strategies.

**Strengths:**

- A strong motivation as there is a clear need for further efficiency improvements
- Introduces a novel rank-adaptive geometry-aware training method that dynamically updates ranks during training
- Proposed to overcome the sensitivity to parameter initialisation and the need for a full-model warm-up phase
- Thorough theoretical analysis, including guarantees on loss descent, convergence, and approximation

**Weaknesses:**

- Claims do not match the results. The abstract says "our training proposal proves to be optimal in locally approximating the original unfactorized dynamics" and while there are guarantees, they are not proven to be optimal.
- Not very clear in many parts. In particular, Section 2.1 is difficult to follow.
- Results seem incomplete. For example, figure 1 does not show compression below 60% and for the proposed method is only shown for 93+% in Figure 1C while the other methods are shown for 60-93%. I could not find any reason for this lack of direct comparison and missing data points.
- According to the plots, the proposed method outperforms the full representation at 96% compression. This is a very surprising finding that requires an in-depth discussion, which is lacking.
- Table 1: The authors say that "TDLRT outperforms the factorization-based and the pruning-based baselines" and bold their method for Alexnet c.r. and Resnet test acc. but according to the same table, baselines actually outperform the proposed method in those metrics.

**Questions:**

- The proposed method appears to outperform the full representation at 96% compression (Figure 1). Can the authors provide an in-depth discussion and analysis of this finding? What factors contribute directly to this performance, and does it align with theoretical expectations?
- Can the authors include data points for the proposed method within the full compression range for a direct comparison with other methods?

**Limitations:**

The paper could benefit from a more detailed discussion on the limitations of the proposed method in different training scenarios and potential strategies to mitigate these limitations.

---

> ### Author Rebuttal · Authors · 2024-08-07
>
> We wish to thank the reviewer for their feedback. Overall, the reviewer highlights key positive aspects of our work, such as the importance of the topic, the novelty of the approach, and the thorough theoretical analysis. The main weaknesses noted are centered on the need for additional runs and a more detailed discussion of results. We are, however, surprised by the perceived impact of these weaknesses, especially in light of the significant strengths identified. The reviewer mentions that our work is "unclear in many parts" and that "Section 2.1 is hard to follow". However, without further details, it is challenging for us to address these points, particularly since clarity has not been a concern for the other reviewers.
> In the following, we address all the identified weaknesses (Ws), as we believe several points merit further clarification. We also answer the raised questions (Qs).
>
> **W1** Thank you for pointing this out. In that sentence of the abstract, our intention was to refer to the optimality of the Dirac-Frenkel variational principle to approximate the dynamics along with the supporting theoretical results of approximation and convergence. However, this statement is not crucial and can be misinterpreted, therefore we will remove it from the abstract.
>
> **W2**
> We recognize that certain sections of the paper, particularly Section 2.1, contain technical details that may require a background in Riemannian optimization and dynamical model-order reduction theory. We have made significant efforts to make this section as clear and accessible as possible, but we understand that some concepts may still be difficult to follow. However, a conference paper is not the ideal format for providing an extensive introduction to these topics.
>
> We would appreciate further clarification on the specific parts of Section 2.1 or other sections that you found unclear. This will help us address your concerns more effectively and improve the paper's readability.
> We aim to clarify the main ideas of Section 2.1:
>
> - Our goal is to derive a low-rank gradient flow that closely approximates the standard full gradient flow. To achieve this, we project the gradient flow onto the tangent bundle as described in Equation (3), which represents the standard gradient flow of Riemannian optimization.
> - However, this equation forms a stiff system that is challenging to solve directly with a discretized scheme, necessitating a small learning rate.
> - By introducing a reparametrization of the weights in Theorem 2.1, we obtain new evolution equations that are well-posed and can be solved with larger learning rates. Despite this improvement, the resulting method remains inefficient as it requires d+1 gradient evaluations.
> - In Corollary 2.2, we demonstrate that the basis computation can be significantly simplified, leading to Algorithm 1, which requires only 2 gradient evaluations instead of d+1.
>
> We point out that we have already attempted to convey this structure and our reasoning at the beginning of Section 2. Please let us know if this explanation clarifies your questions, and if not, which specific parts remain unclear.
>
>
> **W3** and **Q2** The reason our method is sometimes only shown for large compression values is that one key feature of our method is that the compression rate is somewhat determined by the method itself, rather than being a user-defined input, as with the fixed-rank baselines. Nonetheless, we have adjusted the truncation tolerance to obtain smaller compression rates and have included additional data points in the attached rebuttal PDF file. We emphasize that methods at compression rates below 60\% are generally less relevant, so we have focused on the range of large compression rates to maintain the focus on the relevant range of compressions and highlight the strengths of our method. However, if the reviewer considers it valuable, we can include additional runs for smaller compression rates in the appendix.
>
> **W4** and **Q1**
> Thank you for your remark. We emphasize that this phenomenon, where networks with a smaller number of parameters outperform their full baselines, is well-documented in the literature, see e.g. [1,2]. One explanation for this behavior is that neural networks are often overparameterized, and parameter reduction or compression can have a regularizing effect enhancing generalization.
>
> Note that in our comparisons to the baseline, we use the same hyperparameters for compression methods and the full-rank baseline. With this approach on VGG16 we indeed obtain better accuracy than the baseline. However, given your comments, we decided to see if we could find better hyperparameters. This gives us results where the baseline outperforms compression methods. We decided to add these results since the overall accuracy is improved this way for the baseline and for all the compression methods.
>
> [1] The lottery ticket hypothesis: Finding sparse, trainable neural networks, 2018
> [2] Snip: Single-shot network pruning based on connection sensitivity, 2018
>
> **W5**
> Thank you for pointing out this oversight. We apologize for the confusion.
>
> Upon reviewing Table 1, we realized that an error occurred when bolding the results for AlexNet's compression rate (c.r.) and ResNet's test accuracy. We were updating the table with new competitor results shortly before submission and inadvertently left the bold formatting incorrect. We will correct this in our revised manuscript to accurately reflect the performance of each method.
>
> Regarding the specific metrics, it is true that while the TT-factorized method achieves a higher compression rate for AlexNet, its accuracy is lower compared to TDLRT, although it still outperforms other methods significantly. For ResNet18, while the Tucker RGD method achieves a slightly higher accuracy (0.04\%), it does so at the cost of a reduced compression rate.
>
> We will ensure that the table accurately represents these findings and provide a more detailed discussion in the text.

---

### Official Review · Reviewer_SSBR · 2024-07-12

**Soundness:** 3
**Presentation:** 2
**Contribution:** 3
**Rating:** 6
**Confidence:** 3

**Summary:**

The authors study the training of layer factorization models to reduce the number of parameters in deep neural networks. They propose a geometric-aware rank-adaptive training strategy to avoid requiring prior knowledge of ranks and the sensitivity to the weight initializations. Their theoretical results show convergence and approximation error guarantees for the method.

**Strengths:**

The proposed method is quite sensible and is accompanied by good theoretical guarantees.

**Weaknesses:**

The empirical evaluation is slightly weak. While the results could convince me it is better than existing methods (as the proposed method is also quite sensible), from the scale of the model used, it is hard to judge whether it could provide good enough performances on larger models, when model compression during training is more needed.

**Questions:**

Is it possible to conduct experiments on larger models or different architectures (like transformers, even a small one can help) to strengthen the empirical evaluation?

**Limitations:**

Yes, the authors discuss about the limitations.

---

> ### Author Rebuttal · Authors · 2024-08-06
>
> We thank the reviewer for their insightful feedback, below we provide our response to the raised weaknesses and questions.
>
> **Q1**
> Thank you for your feedback regarding the empirical evaluation. In response to your suggestion, we have conducted several new experiments to strengthen our empirical evaluation.
>
> Specifically, we have focused on parameter-efficient fine-tuning using low-rank adapters (LoRA), as suggested by reviewer vGH1. This approach allows us to present additional experimental evidence that complements our previous results. Fine-tuning, rather than compressing a model from scratch during training, enables us to experiment with larger networks while remaining within our time and resource constraints.
>
> In particular, we have tested the proposed training model on larger architectures such as DebertaV3 on Glue benchmark and Stable Diffusion (v1.4) dreambooth. In DebertaV3, we applied our method to matrix layers by interpreting matrices as order-2 tensors. For Stable Diffusion we applied our method to tensor layers in the Unet, while standard LoRA convolution works essentially by reshaping the tensor into a matrix.
>
> These new experiments provide further validation of our method’s performance and demonstrate its applicability to larger models and diverse architectures. We believe these results strengthen the empirical evaluation and could address your concerns about scalability.
>
> | GLUE test case| LoRA - rank 8, 1.33M params | Ours - 0.9 M params |
> |-----------------------|---------------|-----------------------------|
> | CoLa (Matt. Corr)     | $0.6759$      | $0.7065$                    |
> | MRPC (acc)            | $0.8971$      | $0.9052$                    |
> | QQP (acc)             | $0.9131$      | $ 0.9215$                   |
> | RTE (acc)             | $0.8535$      | $0.8713$                    |
> | SST2 (acc)            | $0.9484$      | $0.9594$                    |
>
>
>  Stable Diffusion  | loss | \# trainable parameters
> ----------------------|---------------|----------------------------------
>  LoRA ($r = 8$)       | $0.260$       | $5$ M
>  LoRA ($r = 5$)       | $0.269$       | $3$ M
>  LoRA ($r = 3$)       | $0.274$       | $1.8$ M
>  ours ($\tau = 0.02$) | $0.2635$    | $1.8$ M
>  ours ($\tau = 0.1$)  | $0.272$       | $1.5$ M

---

> > ### Comment · Reviewer_SSBR · 2024-08-11
> >
> > Thanks for the new experiment results. I am raising my score to 6.

---

> > > ### Author Response · Authors · 2024-08-12
> > >
> > > We thank again the reviewer for the feedback.

---

### Official Review · Reviewer_TTwz · 2024-08-01

**Soundness:** 4
**Presentation:** 3
**Contribution:** 3
**Rating:** 7
**Confidence:** 4

**Summary:**

Reducing the size of neural networks is an important problem for reducing the cost, memory usage, and even inference time. Many works focus on reducing the size after the training phase and use techniques such as sparsification and quantization. This paper, on the other hand, focuses on reducing the size by representing the tensor corresponding to the layers of the network as Tucker decompositions. Perhaps the more important aspect is that the paper considers dynamically changing the Tucker rank while training the model. In other words, the size reduction is simultaneous with training and not after the training is performed.

It is argued that due to instability in the gradients, it is required to adopt a geometry-aware training strategy. Essentially, the strategy is to do HOOI based on the gradient of factor matrices. Then, compute a new tensor based on the new factor matrices and update the core tensor based on the gradient of this new tensor. Therefore, for each iteration, two passes are required to update the components: one pass to compute the gradients of factor matrices and one pass to compute the gradient of the core tensor.

The paper presents theoretical results about the convergence and reduction of loss. In addition, it presents a sizable empirical study that shows favorable results for the proposed approach. The approach outperforms a variety of factorization and pruning methods in terms test accuracy and compression rate.

The paper is generally well-written and provides an appropriate method for a very important problem.

I think a relevant paper to discuss is [1], which gives an algorithm to compute the approximately optimal Tucker rank when a size constraint on the size of the Tucker decomposition is given. This could replace the approach based on the tolerance parameter $\tau$ when a hard size constraint is given. It would be interesting to investigate the interaction between that algorithm and the approach presented in this paper.

[1] Ghadiri, Mehrdad, Matthew Fahrbach, Gang Fu, and Vahab Mirrokni. "Approximately optimal core shapes for tensor decompositions." In International Conference on Machine Learning, pp. 11237-11254. PMLR, 2023.

**Strengths:**

The paper is generally well-written and provides an appropriate method for a very important problem.

**Weaknesses:**

-

**Questions:**

The results show that Tucker decomposition works better than CP and tensor-train. Is there any intuition for why this is the case? Do you expect Tucker to be better than any other tensor network, or could decompositions like hierarchical Tucker do better? Hierarchical Tucker could be preferable because of the smaller number of required parameters.

**Limitations:**

I don't see any direct potential negative societal impact.

---

> ### Author Rebuttal · Authors · 2024-08-06
>
> We thank the reviewer for their feedback and appreciation of the work. Below we provide our response to the raised questions.
>
>
> **Q1.** We appreciate this insightful question. Selecting the appropriate tensor structure a-priori is indeed challenging, as it depends heavily on the specific tensor structures in the neural network and the data used. As shown in Table 1 of our main paper, when using standard direct training of the factors, there is no definitive "winner" among Tucker-factorized, Matrix-factorized, TT-factorized, and CP-factorized approaches across the three test problems. However, when employing our proposed geometry-aware TDRLT scheme for training, we observe notably improved performance.
>
> Therefore, we believe that Tucker's superior performance is primarily due to the training method rather than the decomposition choice alone. While extending this training strategy to CP is challenging due to the absence of Riemannian geometry, we believe that further research into extending the proposed geometry-aware training to TT, HT, and other tree tensor network structures would be highly relevant.
>
>
> For what concerns the comment in the "Summary" section, we thank the reviewer for bringing this paper to our attention. We agree that the algorithm presented in [1] offers an interesting alternative approach to consider. In particular, their method is well-suited for scenarios where there is a fixed memory budget, such as training on resource-constrained edge devices. In such cases, the objective shifts from "finding the best accuracy-compression trade-off", which is the focus of our method, to "finding the best-performing Tucker decomposition given memory constraints", where the approach from [1] could be highly beneficial.
>
> To explore this aspect further, it would be useful to extend the method in [1], which currently addresses a single Tucker tensor, to an approach that optimizes the shapes of all tensor-valued layers within the neural network given a global network-wise memory constraint. This is an intriguing research direction that could significantly enhance the applicability of tensor decompositions in constrained environments.
>
> We will certainly include a discussion of this paper in our revised manuscript, highlighting it as a potential alternative approach to our proposed method. This addition will provide readers with a broader perspective on the available techniques and their respective advantages and limitations.

---

> > ### Comment · Reviewer_TTwz · 2024-08-11
> > **Reply to authors' rebuttal**
> >
> > Thank you for answering my questions. I read other reviews and the responses and have decided to keep my score at 7.

---

> > > ### Author Response · Authors · 2024-08-12
> > >
> > > We thank again the reviewer for the feedback.

---

### Author Rebuttal · Authors · 2024-08-07

We would like to thank all the reviewers for their insightful feedback. We have considered each comment and have made several improvements as a result. Below, we address and review the key points raised by the reviewers:

1. **Empirical evaluation**:
Most reviewers commented that while our work presents a promising method with strong theoretical guarantees, additional benchmarks would strengthen the empirical evaluation. Reviewer SSBR and Reviewer Rp73 requested experiments with transformer architectures, while Reviewer vGH1 suggested results using LoRA and U-Net.
In response, we have conducted new experiments that include fine-tuning results via LoRA adapters on DebertaV3 (up to $\sim$ 1.5M parameters) and Stable Diffusion (up to $\sim$ 5M parameters). The results can be found in tables 1 and 2 in the rebuttal pdf file. These new empirical evaluations show that our proposed approach outperforms the LoRA baseline, demonstrating the effectiveness of our method on a broader range of architectures. We thank the reviewers for their suggestions, which have significantly enhanced the quality and robustness of our work.

2. **Clarity and presentation**:
Reviewers were divided on the clarity of our presentation. While Reviewers TTwz, vGH1, and Rp73 found the paper clearly written and easy to follow, Reviewer jjrQ expressed that the paper was difficult to follow and unclear in parts.
To address this, we have done our best to provide explanations addressing specific questions and concerns, aiming to make our presentation more clear without sacrificing the necessary technical rigor. We will do our best to review the manuscript to improve clarity and readability in light of this feedback.

---

### Decision · Program_Chairs · 2024-09-25

**Decision:**

Accept (poster)

**Comment:**

As models continue to grow in size, it is crucial to explore more parameter-efficient alternatives. This paper examines a "parameter saving" technique, namely imposing low-rank constraints. Low-rank constraints have proven effective even for modern transformers. What distinguishes this paper is its combination of theoretical insights and practical aspects. On the theoretical side, the paper analyzes the local convergence of gradient-based optimization under specific low-rank constraints. On the practical side, it proposes adapting these constraints to address optimization challenges, particularly focusing on the geometric aspects of the decomposition. The paper targets an important practical topic and addresses it both theoretically and empirically, I therefore recommend accepting the paper.

There is a substantial body of literature on applying constraints to neural network weight matrices to enhance optimization performance, including constraints based on the Stiefel manifold. The paper could benefit from reviewing this existing research.